DOI: 10.1038/s41467-017-01704-z | **OPEN**

# AIP limits neurotransmitter release by inhibiting calcium bursts from the ryanodine receptor

Bojun Chen[1], Ping Liu[1], Edward J. Hujber[2,3], Yan Li[1], Erik M. Jorgensen[2,3] & Zhao-Wen Wang[1]

Pituitary tumors are frequently associated with mutations in the *AIP* gene and are sometimes associated with hypersecretion of growth hormone. It is unclear whether other factors besides an enlarged pituitary contribute to the hypersecretion. In a genetic screen for suppressors of reduced neurotransmitter release, we identified a mutation in *Caenorhabditis elegans* AIPR-1 (AIP-related-1), which causes profound increases in evoked and spontaneous neurotransmitter release, a high frequency of spontaneous calcium transients in motor neurons and an enlarged readily releasable pool of vesicles. Calcium bursts and hypersecretion are reversed by mutations in the ryanodine receptor but not in the voltage-gated calcium channel, indicating that these phenotypes are caused by a leaky ryanodine receptor. AIPR-1 is physically associated with the ryanodine receptor at synapses. Finally, the phenotypes in *aipr-1* mutants can be rescued by presynaptic expression of mouse AIP, demonstrating that a conserved function of AIP proteins is to inhibit calcium release from ryanodine receptors.

[1] Department of Neuroscience, University of Connecticut Health Center, Farmington, CT 06030, USA. [2] Department of Biology, University of Utah, Salt Lake City, UT 84112, USA. [3] Howard Hughes Medical Institute, University of Utah, Salt Lake City, UT 84112, USA. Correspondence and requests for materials should be addressed to Z.-W.W. (email: zwwang@uchc.edu)

Calcium has pivotal roles in triggering neurotransmitter release. Calcium influx through voltage-gated calcium channels is the main source of calcium driving synaptic vesicle fusion. In addition, smooth endoplasmic reticulum (ER) extends into axon terminals and ryanodine receptor-mediated calcium release from these internal stores regulates synaptic activity[1–4]. Presynaptic ryanodine receptors release calcium in a calcium-dependent manner and can shape synaptic transmission[5]. Specifically, they may enhance spontaneous and evoked neurotransmitter release, promote synchrony of vesicle fusion, and contribute to short-term synaptic plasticity[5–8]. Increased calcium release from ryanodine receptors is considered to be a central factor in the pathogenesis of Alzheimer's disease[9,10]. Thus, identifying molecules that inhibit the ryanodine receptor is important for understanding the mechanisms of both physiological control of synaptic function and diseases associated with leaky RyRs.

The ryanodine receptor is a tetramer of ryanodine receptor proteins (RyR). Human and mouse each have three genes encoding different isoforms of ryanodine receptor proteins: RyR1, RyR2, and RyR3, which display both tissue-specific and overlapping expression patterns. RyR1 and RyR2 are expressed in skeletal and cardiac muscles, respectively, and all three isoforms are expressed in the brain[11]. The physiological functions of ryanodine receptors depend on their interactions with a variety of other proteins. For example, calcium release from ryanodine receptors is modulated by their association with calstabin 1 (FKBP12) and calstabin 2 (FKBP12.6)[11,12], which are immunophilins containing a prolyl isomerase domain with enzyme activity[13]. Dissociation of calstabins from the ryanodine receptor causes increased open probability of the channel[14,15]. In mice with calstabin 2 mutations, leaky ryanodine receptors may cause cognitive dysfunction[15,16] and fatal cardiac arrhythmia[14]. Treatment with a drug that enhances the binding of calstabin 2 to ryanodine receptors can prevent or attenuate seizures, cardiac arrhythmias, and sudden cardiac death caused by mutations of RyR2[17]. The inhibition of ryanodine receptors by calstabins appears to be a conserved mechanism, because ryanodine receptors are similarly regulated by a homolog of calstabins (FKB-2) in *C. elegans*[18].

AIP (aryl hydrocarbon receptor-interacting protein) belongs to the TPR (tetratricopeptide repeat) family of co-chaperones (Fig. 1a). It contains a prolyl-isomerase-like domain on the N-terminal, and a TPR domain and a helical extension to the TPR domain called the "α-7 helix" on the C-terminal[19] (Fig. 1c). Although the prolyl isomerase domain of AIP resembles those of calstabins, it does not have the enzyme activity[20]. The TPR domain contains three TPR motifs[20], which binds the C-terminal residues of the chaperone HSP90[21]. AIP is a suppressor of pituitary tumors[22]. Affected individuals are heterozygous for the mutant allele; the tumors are usually homozygous due to loss of heterozygosity in the pituitary gland in both humans and mice[22–

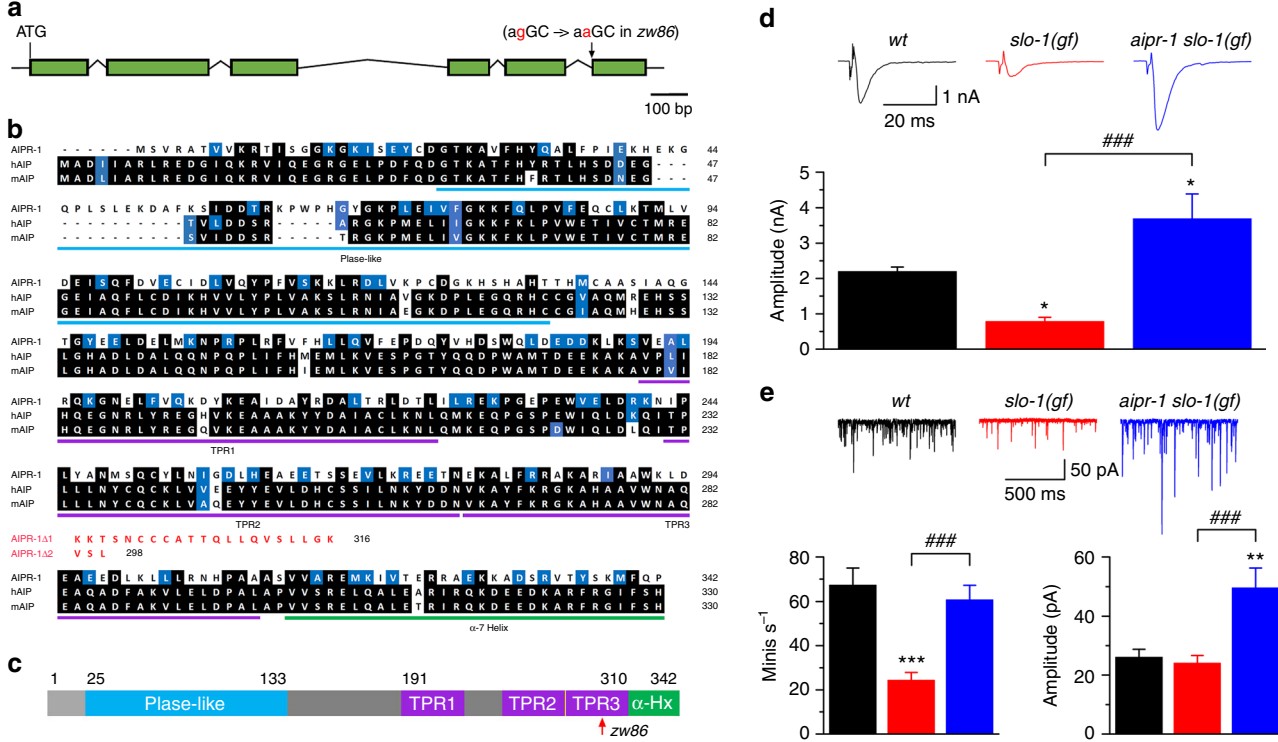

**Fig. 1** The effects of *aipr-1(zw86)* mutation on synaptic phenotypes of *slo-1* gain-of-function (*gf*). **a** Schematic diagram showing the exon and intron organization of *aipr-1* (GenBank: NP_495339.1). zw86 is a G to A transition in the splice acceptor site before the last exon. **b** Alignment of amino acid sequences between AIPR-1, human AIP (hAIP) (GenBank: ACN38897.1), and mouse AIP (mAIP) (GenBank: AAH75614.1). AIPR-1 is 35% identical to hAIP. Identical residues are highlighted in black, whereas similar ones (in size, acidity, or polarity) in blue. The prolyl *cis–trans* isomerase-like domain (Plase-like), tetratricopeptide repeat (TPR) motifs, and the C-terminal α-7 helix (α-Hx) of the TPR repeat structure are underlined. *aipr-1(zw86)* does not make wild-type AIPR-1 but may produce two alternative isoforms truncated after glutamate (E) 295 (AIPR-1Δ1 and AIPR-1Δ2) with additional out-of-frame amino acid residues. **c** Diagram of AIPR-1 domain structure. The red arrow indicates the location of AIPR-1 truncation in zw86. **d** Comparison of evoked current amplitude among wild-type (*wt*), *slo-1(gf)*, and the *slo-1(gf) aipr-1(zw86)* double mutant. *n* = 8 in all groups. **e** Comparison of the frequency and mean amplitude of spontaneous minis among the three groups. *n* = 8 in all groups. Data are shown as mean ± SEM. *$p < 0.05$, **$p < 0.01$, ***$p < 0.001$ compared with *wt*; ### $p < 0.001$ compared between with *slo-1(gf)* (one-way ANOVA followed by Tukey's *post-hoc* test)

[24]. The aryl hydrocarbon receptor-cAMP-phosphodiesterase pathway appears to be important for why AIP mutations cause pituitary tumors[25]. Patients with pituitary tumors including those caused by mutations of the *AIP* gene frequently exhibit acromegaly and gigantism due to hypersecretion of growth hormone[22,26,27]. It is unknown whether any other factors besides the increased pituitary size have a role in the hypersecretion of growth hormone. Although at least 20 interacting proteins have been identified for AIP, none suggest an obvious direct mechanism for growth hormone hypersecretion[20]. In addition, AIP is abundantly expressed in the human brain (http://human.brain-map.org/) but little is known about its physiological roles in the nervous system.

Here we demonstrate that AIPR-1 (AIP-related 1), the worm ortholog of human AIP, is an inhibitor of presynaptic ryanodine receptors. We find that a mutation in AIPR-1 results in increased release of neurotransmitter and this effect is occluded by a loss-of-function mutation of the ryanodine receptor but not of a presynaptic voltage-gated calcium channel. In addition, the AIPR-1 mutation causes increases in synaptic vesicle number and the readily releasable pool size through a ryanodine receptor-independent pathway. This study identifies AIPR-1/AIP as a critical regulator of neurotransmitter release.

## Results

### AIPR-1 is a potent inhibitor of synaptic transmission.

We identified *aipr-1* in a forward genetic screen for mutants that increased neurotransmitter release in a sensitized background. In a worm strain expressing a hyperactive BK channel SLO-1, a calcium-activated potassium channel that hyperpolarizes the synaptic terminal and limits exocytosis, neurotransmitter release is greatly reduced, causing a lethargic phenotype[28]. We screened for suppressors of the sluggish phenotype and identified the mutation *zw86*, which ameliorated the locomotion defect of the *slo-1(gf)* strain. Mapping, sequencing, and rescue experiments established that the *zw86* mutation is an allele of the gene *aipr-1*, the worm ortholog of mammalian AIP (Fig. 1a–c). *zw86* is a G to A transition in the splice acceptor site before the last exon (Fig. 1a). Sequencing of *aipr-1(zw86)* complementary DNA revealed that the mutant does not make wild-type AIPR-1 but may produce two alternative isoforms with frame shifts (Fig. 1b). *aipr-1* is the third gene in an operon of four genes, including *epg-5* (C56C10.12 + C56C10.11), *dnj-8*, *aipr-1*, and C56C10.9. The *zw86* mutation did not affect mRNA levels of the other three genes in the operon (Supplementary Fig. 1). The *aipr-1* mutant is hyperactive; quantification with an automated worm tracker[29] showed that locomotion speed of the mutant doubled compared to the wild type (Supplementary Fig. 2).

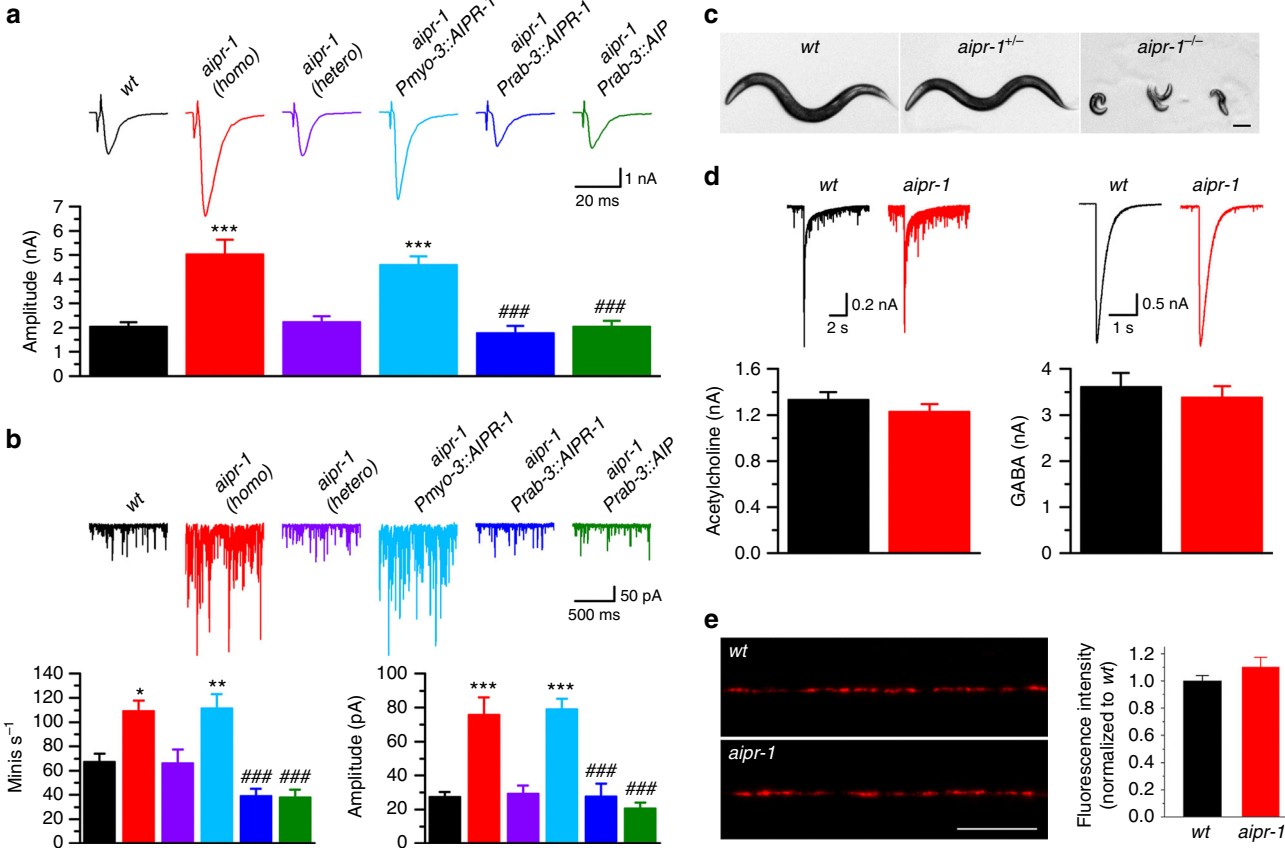

**Fig. 2** Augmented synaptic transmission in *aipr-1(zw86)*. **a**, **b** Comparison of evoked currents **a** and spontaneous miniature currents (minis) **b** at the neuromuscular junction among wild type (*wt*) (*n* = 7), homozygous *aipr-1(zw86)* (*n* = 7), heterozygous *aipr-1(zw86)* (*n* = 7), and *aipr-1(zw86)* rescued either in neurons (P*rab-3::AIRP-1*, *n* = 7; P*rab-3::AIP*, *n* = 9) or body-wall muscle cells (P*myo-3::AIPR-1*, *n* = 7). **c** Homozygous *aipr-1* knockout (*aipr-1^{-/-}*) worms arrest at early larval stages. The knockout results from a 2 bp deletion in the first exon of *aipr-1*. Scale bar, 100 μm. **d** Comparison of muscle inward current in response to exogenous acetylcholine (100 μM) and GABA (100 μM) between *wt* (ACh, *n* = 7; GABA, *n* = 7) and *aipr-1(zw86)* (ACh, *n* = 8; GABA, *n* = 8). Data are shown as mean ± SEM. *$p < 0.05$, **$p < 0.01$, ***$p < 0.001$ compared with *wt*; ###$p < 0.001$ compared with *aipr-1(zw86)* (one-way ANOVA followed by Tukey's *post-hoc* test). **e** tagRFP-labeled muscle acetylcholine receptor (UNC-29) distribution and fluorescence intensity are normal in *aipr-1(zw86)* (*wt*, *n* = 20; *aipr-1*, *n* = 23, unpaired *t*-test). Scale bar, 10 μm

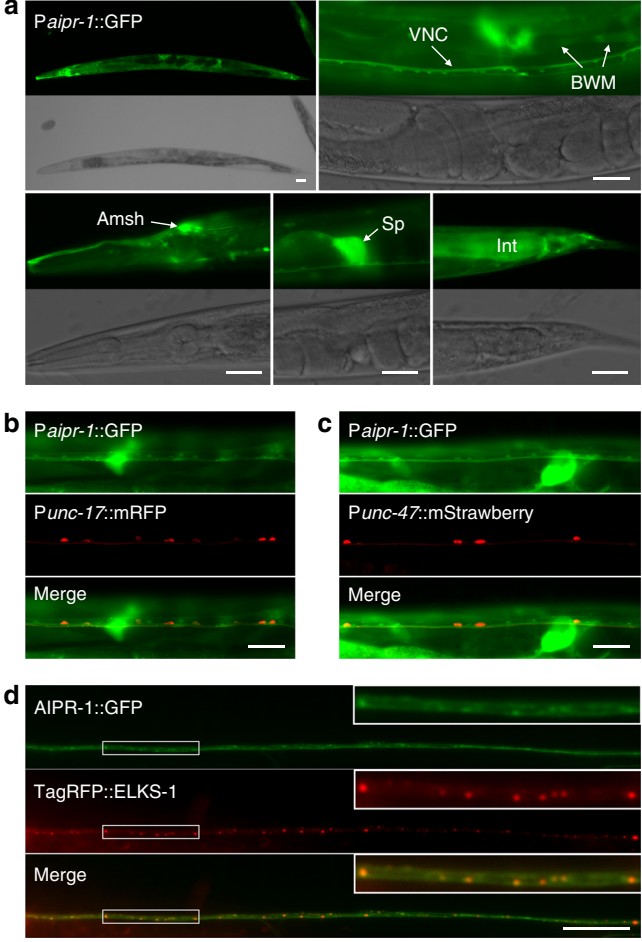

**Fig. 3** Expression and subcellular localization of AIPR-1. **a** A GFP reporter, expressed under the control of genomic DNA upstream of the *aipr-1* initiation site, showed expression in many cell types, such as motor neurons along the ventral nerve cord (VNC), body-wall muscle cells (BWM), amphid sheath cells (Amsh), spermatheca (Sp), and the intestine (Int). **b**, **c** AIPR-1 is expressed in all acetylcholine and GABA motor neurons in the ventral nerve cord. Acetylcholine and GABA motor neurons were labeled by expressing mRFP and mStrawberry under the control of P*unc-17* and P*unc-47*, respectively. Images were taken from a segment anterior **b** and posterior **c** to the vulva. **d**, AIPR-1::GFP displayed both diffuse and punctate distribution along the dorsal nerve cord, and the GFP puncta colocalized with the presynaptic marker TagRFP::ELKS-1. Images were taken from a dorsal segment anterior to the vulva. Scale bars, 20 μm **a–c** and 10 μm **d, e**

The increased activity observed in the *slo-1(gf) aipr-1(zw86)* double mutant suggests that a potential physiological function of AIPR-1 is to limit synaptic activity. Indeed, electrophysiological recordings from the neuromuscular junction demonstrated that the *zw86* mutation increases synaptic transmission in *slo-1(gf)* mutants. The frequency and mean amplitude of spontaneous miniature postsynaptic currents ('minis') and the amplitude of evoked postsynaptic currents are increased in the *aipr-1 slo-1(gf)* double mutant compared with *slo-1(gf)* single mutants (Fig. 1d, e).

The single mutant *aipr-1(zw86)*, when outcrossed from *slo-1(gf)*, exhibits a doubled evoked response (wt $2.1 \pm 0.2$ nA; *aipr-1* $4.9 \pm 0.5$ nA), an increased frequency of spontaneous minis (wt $66.9 \pm 11.3 s^{-1}$; *aipr-1* $107.2 \pm 7.4 s^{-1}$), and an increased amplitude of minis (wt $27.6 \pm 3.3$ pA; *aipr-1* $74.4 \pm 8.7$ pA) compared with the wild type in the presence of 5 mM $[Ca^{2+}]_o$ (Fig. 2a, b). Both evoked responses and minis were normal in heterozygous

*aipr-1(zw86)* (Fig. 2a, b), suggesting that the mutation is fully recessive. Despite being isolated as a suppressor of a hyperactive SLO-1 potassium channel, AIPR-1 does not act in the same pathway as SLO-1. If AIPR-1 were to inhibit neurotransmission by activating SLO-1, then *aipr-1* mutants should have resembled *slo-1* loss-of-function (*lf*) mutants but should not exhibit a more severe phenotype. However, synaptic phenotypes of *aipr-1(zw86)* are much more severe than those of *slo-1(lf)*. Although the mutation of *aipr-1* causes large increases in evoked responses and in the frequency and mean amplitude of minis at both 0.5 and 5.0 mM $[Ca^{2+}]_o$, *slo-1(lf)* shows only smaller increases in evoked responses and the frequency of minis at the lower $[Ca^{2+}]_o$ (Supplementary Fig. 3a, b). Moreover, *aipr-1(zw86)* does not alter either SLO-1 expression or subcellular localization (Supplementary Fig. 3c-e), unlike other mutants that suppress SLO-1(*gf*)[28,30,31]. Thus, AIPR-1 inhibits neurotransmission through a different pathway than through the SLO-1 potassium channel.

**Worm and mammalian AIP act presynaptically.** Similar to the human protein, AIPR-1 is composed of an FKBP-type prolyl isomerase domain, a TPR domain with three TPR motifs and the associated "α-7" helix (Fig. 1c). cDNA sequencing indicates that the two incorrectly spliced isoforms with frame shifts disrupt the last TPR motif and the α-7 helix (Fig. 1b). Similarly, most mutations of *AIP* in human patients disrupt or delete the last TPR motif and the α-7 helix[20,24]. We generated a deletion allele of *aipr-1* using CRISPR/Cas9; *zw90* homozygous animals arrest at early larval stages (Fig. 2c), which suggest that *aipr-1(zw86)* is a hypomorph. Similarly, knockouts of the *AIP* gene in mouse are embryonic lethal[32].

To test whether mammalian AIP can substitute for AIPR-1 function, we expressed mouse AIP in *aipr-1(zw86)* neurons under the control of the *rab-3* promoter (P*rab-3*). Mouse AIP rescued the synaptic phenotypes (Fig. 2a, b) (evoked: *aipr-1* $4.9 \pm 0.5$ nA, *aipr-1* P*rab-3*::AIPR-1 $1.8 \pm 0.3$ pA, *aipr-1* P*rab-3*:AIP $2.0 \pm 0.2$ pA; minis: *aipr-1* $74.4 \pm 8.7$ pA, *aipr-1* P*rab-3*:AIPR-1 $27.7 \pm 7.5$ pA, *aipr-1* P*rab-3*:AIP $20.7 \pm 3.2$ pA; mini frequency: *aipr-1* $107.2 \pm 7.4 s^{-1}$, *aipr-1* P*rab-3*:AIPR-1 $39.5 \pm 5.5 s^{-1}$, aipr-1 P*rab-3*:AIP $38.3 \pm 6.0 s^{-1}$). The rescue of *aipr-1(zw86)* synaptic phenotypes by mouse AIP suggests that the proteins have similar roles in worms and mammals. Furthermore, rescue in neurons suggests that AIPR-1 acts presynaptically rather than in the postsynaptic muscles. Consistent with this conclusion, expression of AIPR-1 in muscles under the control of the *myo-3* promoter (P*myo-3*) did not rescue *aipr-1* mutants (evoked: $4.6 \pm 0.3$ pA; minis: $79.3 \pm 5.9$ pA; mini frequency $111.8 \pm 11.2 s^{-1}$ (Fig. 2a, b). Furthermore, direct application of either acetylcholine or γ-aminobutyric acid (GABA) onto muscle produces normal currents in *aipr-1(zw86)* (Fig. 2d), and the expression of a tagRFP-tagged UNC-29[33], which is a key subunit of muscle acetylcholine receptor[34], was similar between the wild type and *aipr-1(zw86)* (Fig. 2e), suggesting that muscle physiology is normal.

The *aipr-1(zw86)* mutants do not exhibit morphological changes in acetylcholine and GABA motor neurons (Supplementary Fig. 4a, b) or an increase in the density of presynaptic sites (Supplementary Fig. 4c, d), suggesting that hypersecretion is caused by a defect in synaptic function rather than development. Because both acetylcholine and GABA neurons synapse onto muscles, we recorded acetylcholine and GABA minis separately. Minis from both neurotransmitters exhibit higher frequencies and larger amplitudes in *aipr-1(zw86)* than in the wild type (Supplementary Fig. 5), indicating that AIPR-1 is not restricted to a particular synaptic type.

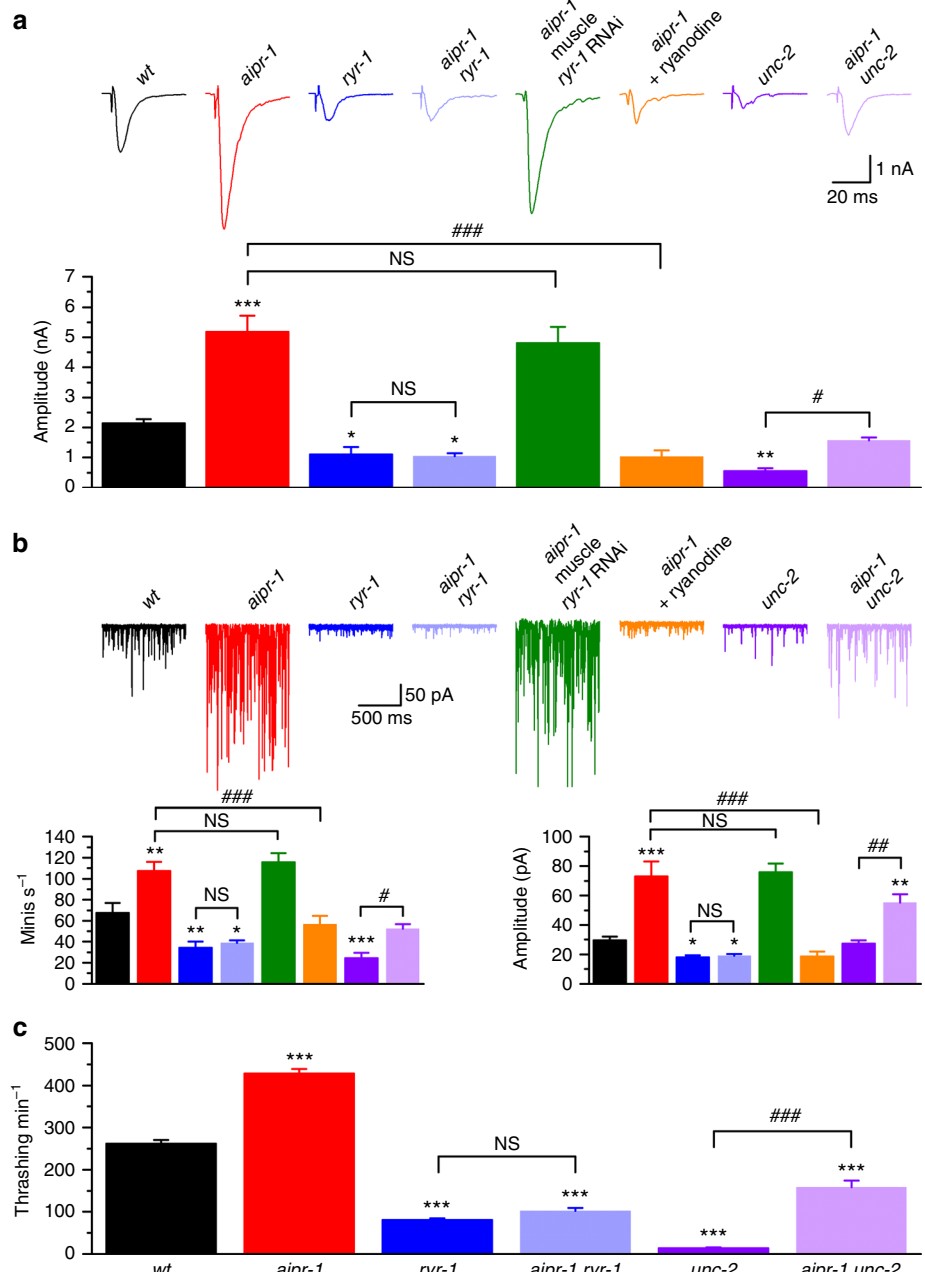

**Fig. 4** Mutation of RYR-1 but not UNC-2 occluded synaptic and behavioral phenotypes of *aipr-1(zw86)*. **a**, **b** Comparison of evoked currents **a** and spontaneous miniature currents (minis) **b** among wild type (*wt*) (*n* = 8), *aipr-1(86)* (*n* = 7), *ryr-1(e540)* (*n* = 7), *aipr-1(zw86) ryr-1(e540)* (*n* = 7), *aipr-1 (zw86)* with muscle *ryr-1* RNAi (*n* = 7), *aipr-1(zw86)* treated with ryanodine (100 μM) (*n* = 8), *unc-2(e55)* (*n* = 9), and *aipr-1(zw86) unc-2(e55)* (*n* = 7). **c** Comparison of thrashing rates among these groups (*n* = 12 or 13). Data are shown as mean ± SEM. *$p < 0.05$, **$p < 0.01$, ***$p < 0.001$ compared with *wt*; #$p < 0.05$, ##$p < 0.01$, ###$p < 0.001$ compared between indicated groups; NS $p > 0.05$ (one-way ANOVA followed by Tukey's *post-hoc* test)

To determine the expression pattern of AIPR-1, we generated a transcriptional reporter in an extrachromosomal array by in vivo recombination. A fosmid (WRM065bd01) containing the entire operon was coinjected with a plasmid containing homologous DNA encompassing 0.5 kb upstream of the *aipr-1* start codon fused to green fluorescent protein (GFP). In transgenic worms, GFP was observed in a variety of cell types, including neurons, body muscles, amphid sheath cells, spermatheca, and the intestine (Fig. 3a). Both acetylcholine and GABA motor neurons express *aipr-1*, as indicated by the co-labeling of these neurons by a red fluorescent protein expressed under the control of cell-specific promoters (Fig. 3b, c). To determine the subcellular localization

of AIPR-1, we tagged the protein with GFP at either the N or C terminus using CRISPR/Cas9. Both strains exhibit wild-type behavior, suggesting that the fusion proteins are functional. Electrophysiological responses were tested in the AIPR-1::GFP strain; evoked responses and minis were normal (Supplementary Fig. 6a, b). However, only very dim GFP fluorescence could be detected in these strains (Supplementary Fig. 6c). The lack of a strong GFP signal in the nerve cords prevented us from analyzing synaptic localization of AIPR-1. To determine AIPR-1 subcellular localization in neurons, we overexpressed AIPR-1::GFP under the control of the panneuronal *rab-3* promoter. The tagged protein

was found throughout axons and was enriched at synapses, as determined by colocalization with a presynaptic marker (Fig. 3d).

**AIPR-1 inhibits calcium bursts from ryanodine receptors**. The electrophysiological data suggest that AIPR-1 normally inhibits synaptic transmission. What then is the molecular target of AIPR-1? The prolyl isomerase-like domains of AIP resemble the prolyl isomerase domains of calstabin 1 and calstabin 2[20]. Because calstabins inhibit ryanodine receptor-mediated $Ca^{2+}$ release from the ER[14,15,35,36], the absence of AIPR-1 might also disinhibit the ryanodine receptor, causing frequent calcium bursts and increased exocytosis. We therefore tested whether AIPR-1 acts via the sole ryanodine receptor of the worm (called UNC-68 or RYR-1, we will use RYR-1 henceforth for clarity). The ryr-1 gene generates transcripts from two promoters (www.wormbase.org). Reporter constructs demonstrated that the upstream promoter is expressed in muscle, whereas the downstream promoter is expressed in neurons (Supplementary Fig. 7). To determine whether the ER extends into synaptic regions, we coexpressed an ER marker (mStrawberry::PISY-1) and a presynaptic marker (GFP::ELKS-1) in motor neurons. We found that the ER marker was distributed throughout the dorsal nerve cord, including synaptic regions (Supplementary Fig. 8), which is consistent with a presynaptic role for RYR-1.

Loss of ryanodine receptor function in both mammals and worms cause synaptic phenotypes opposite to those in aipr-1 (zw86)[6–8,37]. Specifically in worms, we see decreases in the amplitude of evoked currents and in the frequency and amplitude of spontaneous minis in mutants of ryr-1[8] (Fig. 4a, b). The profound increases in evoked and spontaneous currents in aipr-1 mutants are completely abolished by ryr-1(e540), a putative ryr-1-null mutant[38], and the double mutant exhibits reduced evoked and spontaneous currents identical to the ryr-1 single mutant (evoked: aipr-1 $5.1 \pm 0.5$ nA; ryr-1 $1.1 \pm 0.2$ nA; aipr-1 ryr-1 $1.1 \pm 0.1$ nA; minis: aipr-1 $73.3 \pm 9.9$ pA; ryr-1 $18.2 \pm 1.1$ pA; aipr-1 ryr-1 $19.2 \pm 0.9$ pA; mini frequency: aipr-1 $107.6 \pm 8.6$ $s^{-1}$; ryr-1 $34.5 \pm 5.8$ $s^{-1}$; aipr-1 ryr-1 $38.9 \pm 2.4$ $s^{-1}$) (Fig. 4a, b). Similarly,

acute inhibition of ryanodine receptors using $100\,\mu M$ ryanodine abolished the increased evoked responses and minis in aipr-1 mutant (Fig. 4a, b), suggesting that the suppression of aipr-1 synaptic phenotypes by ryr-1 mutation is not due to a developmental defect. Knockdown of ryr-1 specifically in muscle had no effect on either evoked responses or minis of aipr-1(zw86) (Fig. 4a, b), suggesting that loss of presynaptic ryr-1 occludes the synaptic phenotypes of the aipr-1 mutant. Another major source of calcium at synapses is the N-type voltage-gated calcium channel ($Ca_V2$) encoded by unc-2[39–42]. Loss-of-function mutations in unc-2 also cause reduced evoked and spontaneous currents. However, in contrast to the aipr-1 ryr-1 mutant, the aipr-1 unc-2 double mutant exhibits increased neurotransmission compared with unc-2 alone (Fig. 4a, b), suggesting that at least part of the AIPR-1 function does not require the N-type calcium channel. Finally, ryr-1 also suppresses the behavioral phenotypes of aipr-1. In solution, aipr-1(zw86) worms show a much higher thrashing rate compared to the wild type and this behavioral phenotype is suppressed by ryr-1 but not unc-2 mutations (Fig. 4c and Supplementary Movies 1-6). The specificity of ryr-1(lf) in suppressing both the synaptic and behavioral phenotypes of aipr-1(zw86) suggests that AIPR-1 is an inhibitor of the RYR-1 channels.

To determine whether AIPR-1 inhibits calcium bursts from ryanodine receptors, we examined calcium transients in motor neurons using the $Ca^{2+}$ indicator GCaMP6f[43]. To knock down AIPR-1 expression, we performed RNA interference (RNAi). RNAi is expected to specifically act on aipr-1 transcripts although we cannot exclude the possibility that RNAi might affect other genes of the operon. To avoid potential complications from increased activity of upstream neurons, we performed motor neuron-specific RNAi for aipr-1, ryr-1, or both. The frequency of $Ca^{2+}$ transients is more than doubled when aipr-1 is knocked down in A-type motor neurons and these frequent calcium bursts are occluded by ryr-1 knockdown (Fig. 5 and Supplementary Movies 7-10). These results suggest that reducing AIPR-1 function results in a "leaky" RYR-1, causing excessive $Ca^{2+}$ release from the ER.

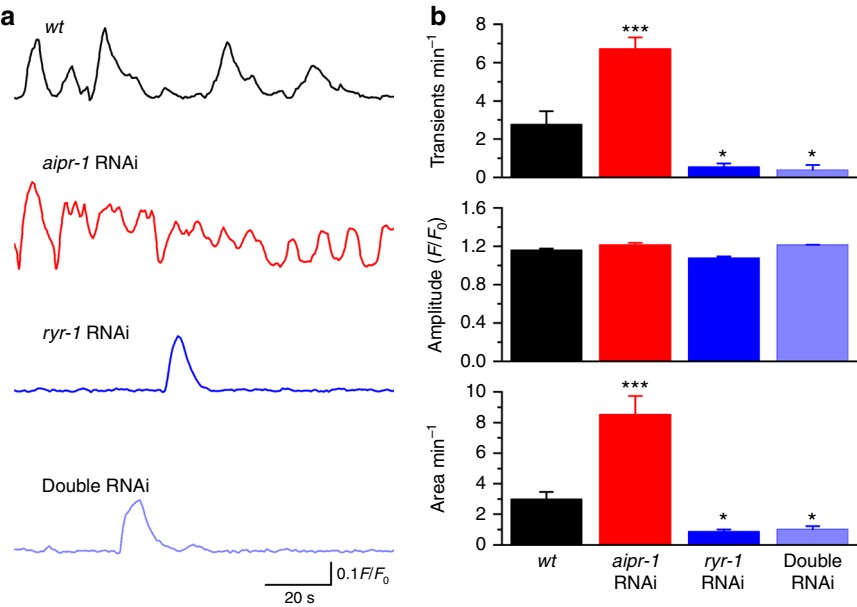

**Fig. 5** AIPR-1 inhibits neuronal $Ca^{2+}$ bursts from the ryanodine receptor. Expression of either aipr-1, ryr-1, or both genes was knocked down in A-type motor neurons using RNAi. **a** Representative traces of $Ca^{2+}$ transients, determined in A-type motor neurons using GCaMP6f. **b** Bar graphs of calcium transient frequencies, amplitudes, and area in the wild type (wt) ($n = 9$), aipr-1 knockdown ($n = 10$), ryr-1 knockdown ($n = 6$), and aipr-1 ryr-1 double knockdown ($n = 6$). Data are shown as mean ± SEM. *$p < 0.05$, ***$p < 0.001$ compared with wt; (one-way ANOVA followed by Tukey's post-hoc test)

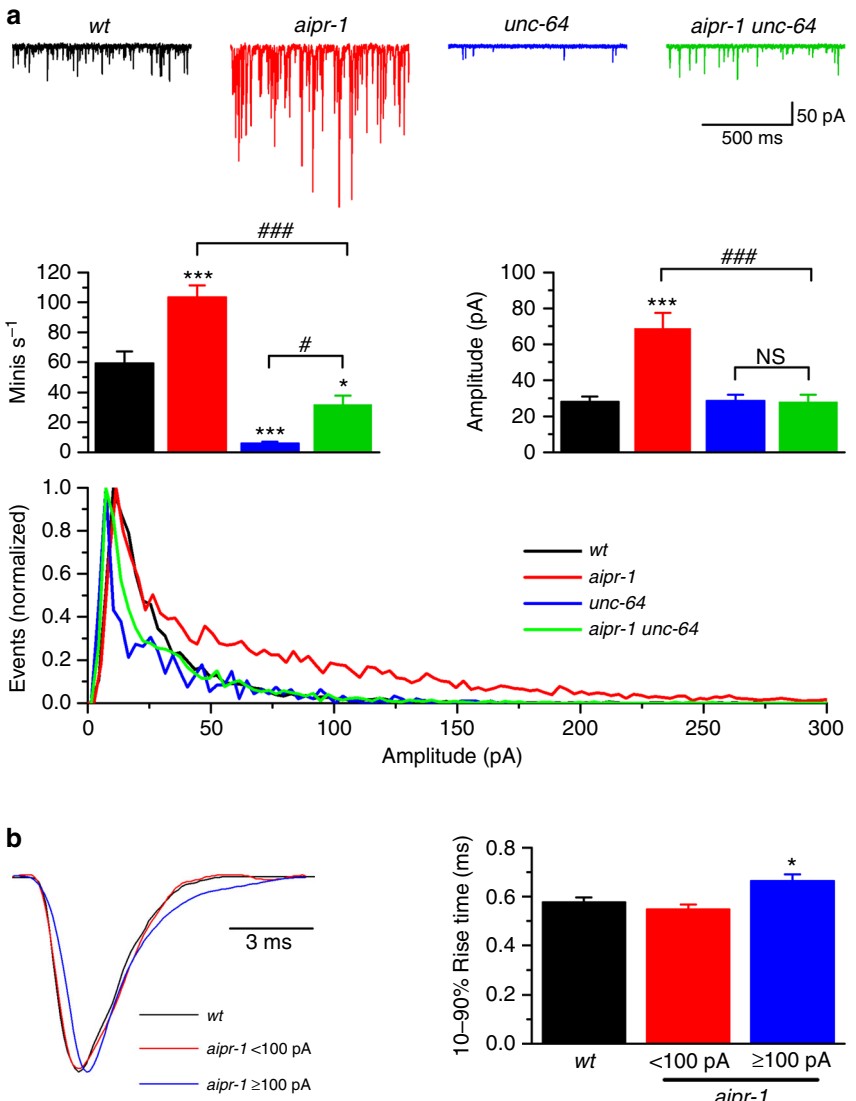

**Fig. 6** AIPR-1 inhibits synchrony of spontaneous release. **a** Mutation of *unc-64* (syntaxin) precluded the enhancing effect of *aipr-1(zw86)* on the amplitude of spontaneous miniature currents (minis). Frequency, mean amplitude, and amplitude distribution of minis are compared among the wild type (*wt*) ($n =$ 9), *aipr-1(zw86)* ($n = 7$), *unc-64(e246)* ($n = 9$), and *aipr-1(zw86) unc-64(e246)* ($n = 8$). It is noteworthy that the *aipr-1* mutation increased large-amplitude events in the amplitude distribution graph and this effect was eliminated by a mutation in *unc-64*. **b** Comparison of 10–90% rise time of minis between wild type (*wt*) and *aipr-1(zw86)*. The sample minis were normalized in amplitude and superimposed at an identical time scale. Data are shown as mean ± SEM. *$p < 0.05$, ***$p < 0.001$ compared with *wt*; #$p < 0.05$, ###$p < 0.001$, and NS ($p > 0.05$) compared between indicated groups (one-way ANOVA followed by Tukey's *post-hoc* test)

Previous studies have shown that activation of presynaptic ryanodine receptors leads to large minis by promoting synchronous multivesicular release[6,7]. If large minis are caused by multivesicular release, rather than increases in vesicle filling or postsynaptic responses, then decreasing release probability using mutations in the SNARE (soluble N-ethylmaleimide-sensitive factor attachment protein receptor) fusion machinery should restore mini amplitude. Syntaxin is encoded by the *unc-64* gene in *C. elegans*[44] and the hypomorphic *unc-64(e246)* mutant exhibits reduced mini frequency[8,45]. In *aipr-1(zw86) unc-64(e246)* double mutant, mini frequency is reduced and mini amplitudes are restored to wild-type levels (Fig. 6a), suggesting that large minis in the *aipr-1* mutant are likely due to multivesicular release. Furthermore, we analyzed the rise times of minis, which are predicted to increase if large minis are due to multivesicular release[6,7]. Our previous study shows that, in wild-type worms, all minis are mono-quantal events with a similar rise time regardless of the amplitude[8]. We divided minis of the *aipr-1* mutant into two categories based on amplitudes ( $< 100$ pA and $\geq 100$ pA), and compared the mean rise times with the wild type. We found that the rise time of large minis in the *aipr-1* mutant was significantly longer than that of wild-type minis (Fig. 6b). Collectively, these observations suggest that the large-amplitude minis in the *aipr-1* mutant are caused by synchronous multivesicular release as a result of elevated $Ca^{2+}$ concentration at the presynaptic terminal, and that the function of AIPR-1 at the synapse is to reduce calcium bursts mediated by the ryanodine receptor.

To determine whether AIPR-1 might physically interact with RYR-1, we performed bimolecular fluorescence complementation (BiFC) assays[46]. The N-terminal portion of YFP (YFPa) was fused to the C terminus of the ryanodine receptor RYR-1 and the C-terminal portion of YFP (YFPc) was fused to the C terminus of AIPR-1 (Fig. 7a). We observed YFP fluorescence when the two

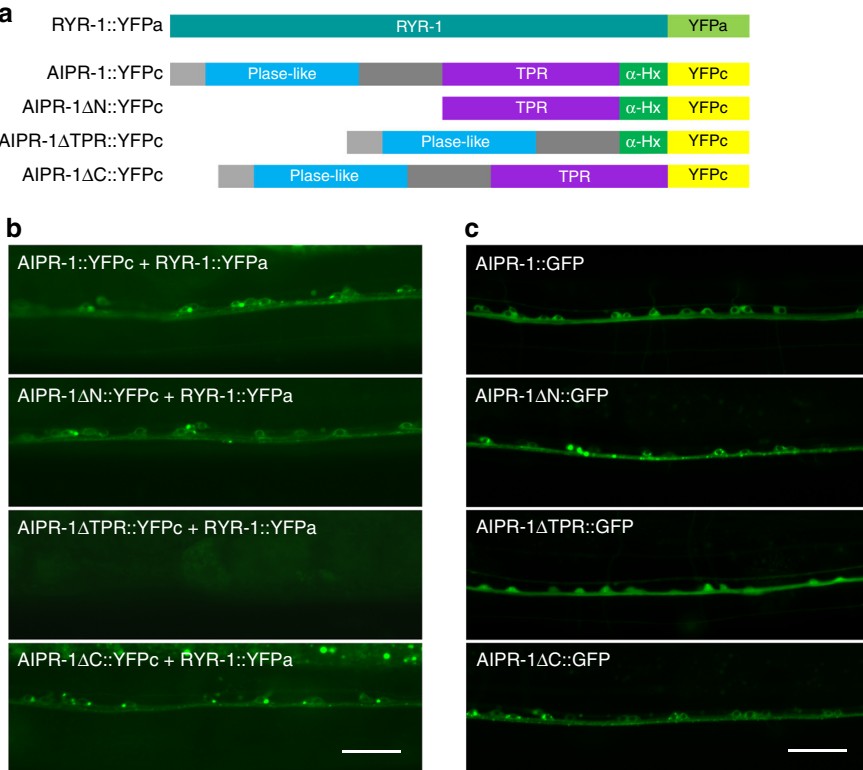

**Fig. 7** Bimolecular fluorescence complementation between AIPR-1 and RYR-1 in vivo. **a** Schematic diagrams showing the various fusion proteins used in the bimolecular fluorescence complementation assays. **b** Deletion of the TPR domain but not any other parts of AIPR-1 prevented AIPR-1::YFPc from reconstituting YFP fluorophore with RYR-1::YFPa in ventral cord motor neurons. **c** GFP-tagged AIPR-1 with the various deletions showed comparable expression levels in ventral cord motor neurons. Scale bars, 20 μm

fusion proteins were coexpressed in neurons (Fig. 7b). YFP fluorescence was also observed when either the prolyl isomerase domain or the C-terminal α-7 helix was deleted (Fig. 7b). However, YFP fluorescence was not detected when the TPR domain was deleted (Fig. 7b). The lack of BiFC with the TPR deletion is unlikely due to a lack of protein expression, because this deletion did not appreciably alter the expression level of a GFP fusion protein (Fig. 7c). These results suggest that AIPR-1 is physically associated with RYR-1 and this association requires the TPR domain. This is consistent with an established role of TPR domains in mediating protein-protein interactions[47] and with the loss-of-function phenotype of zw86, which disrupts the last TPR motif (Fig. 1).

***aipr-1* mutants accumulate vesicles independent of RYR-1.** To determine whether changes in vesicle number or size are contributing factors to *aipr-1(zw86)* synaptic phenotypes, we performed ultrastructural analyses and quantified the number and size of synaptic vesicles at GABA and acetylcholine synapses. The *aipr-1(zw86)* mutant exhibits a nearly twofold increase in the number of synaptic vesicles (GABA neurons, Fig. 8a, b and Supplementary Fig. 10d'; ACh neurons, Supplementary Figs. 9 and 10a,d). Docked vesicle numbers also trend upward, although to a lesser degree (GABA neurons, Fig. 8c and Supplementary Fig. 10c'; ACh neurons, Supplementary Fig. 10b,c). Interestingly, *ryr-1(lf)* does not mask these effects (total vesicles: *wt* $23.1 \pm 2.1$; *aipr-1* $40.1 \pm 2.2$; *ryr-1* $27.2 \pm 2.8$; *aipr-1 ryr-1* $43.4 \pm 2.6$; docked vesicles: *wt* $0.8 \pm 0.2$; *aipr-1* $1.2 \pm 0.2$; *ryr-1* $0.8 \pm 0.2$; *aipr-1 ryr-1* $0.9 \pm 0.1$). These observations suggest that, besides regulating $Ca^{2+}$ release by RYR-1, AIPR-1 has effects on vesicle biogenesis.

To determine whether these docked vesicles were primed, we stimulated release using a hypertonic sucrose solution (Fig. 8d). This $Ca^{2+}$-independent response assesses the size of the readily releasable pool by driving exocytosis of docked vesicles[48]. The readily releasable pool is increased in *aipr-1* using this assay as well, the sucrose response was increased ~ 3-fold in *aipr-1(zw86)* compared with the wild type (*wt* $40.5 \pm 8.5$ pC; *aipr-1* $148.6 \pm 16.6$ pC). Similar to the increase in total vesicles, the increase in the readily releasable pool is not caused by dysregulation of the ryanodine receptor: the sucrose response was increased by more than twofold in the *aipr-1 ryr-1* double mutant compared with *ryr-1* (*ryr-1* $42.9 \pm 8.9$ pC; *aipr-1 ryr-1* $97.3 \pm 16.8$ pC). The enhanced sucrose response in *aipr-1* mutant was completely abolished by a mutation of UNC-13/Munc-13, which has a pivotal role in synaptic vesicle priming[49,50], suggesting that the increase in primed vesicles does not bypass the normal mechanisms for docking and priming.

## Discussion
The striking increase in neurotransmitter release seen in the *aipr-1* mutant demonstrates that the AIPR-1/AIP protein is a major inhibitor of neurotransmission. Loss of AIPR-1/AIP results in increases in several aspects of neurotransmitter release, including evoked responses, spontaneous vesicle fusion, and current amplitude of spontaneous events. These effects can be attributed to disinhibition of the ryanodine receptor and the concomitant increase in calcium release from internal stores. In addition to the ryanodine receptor, AIP must act on at least one other target that limits synaptic vesicle number at neuromuscular junctions. Given that the mouse AIP can rescue the synaptic phenotypes of *aipr-1* mutant, the regulation of ryanodine receptors by AIPR-1/AIP is likely conserved between worms and mammals.

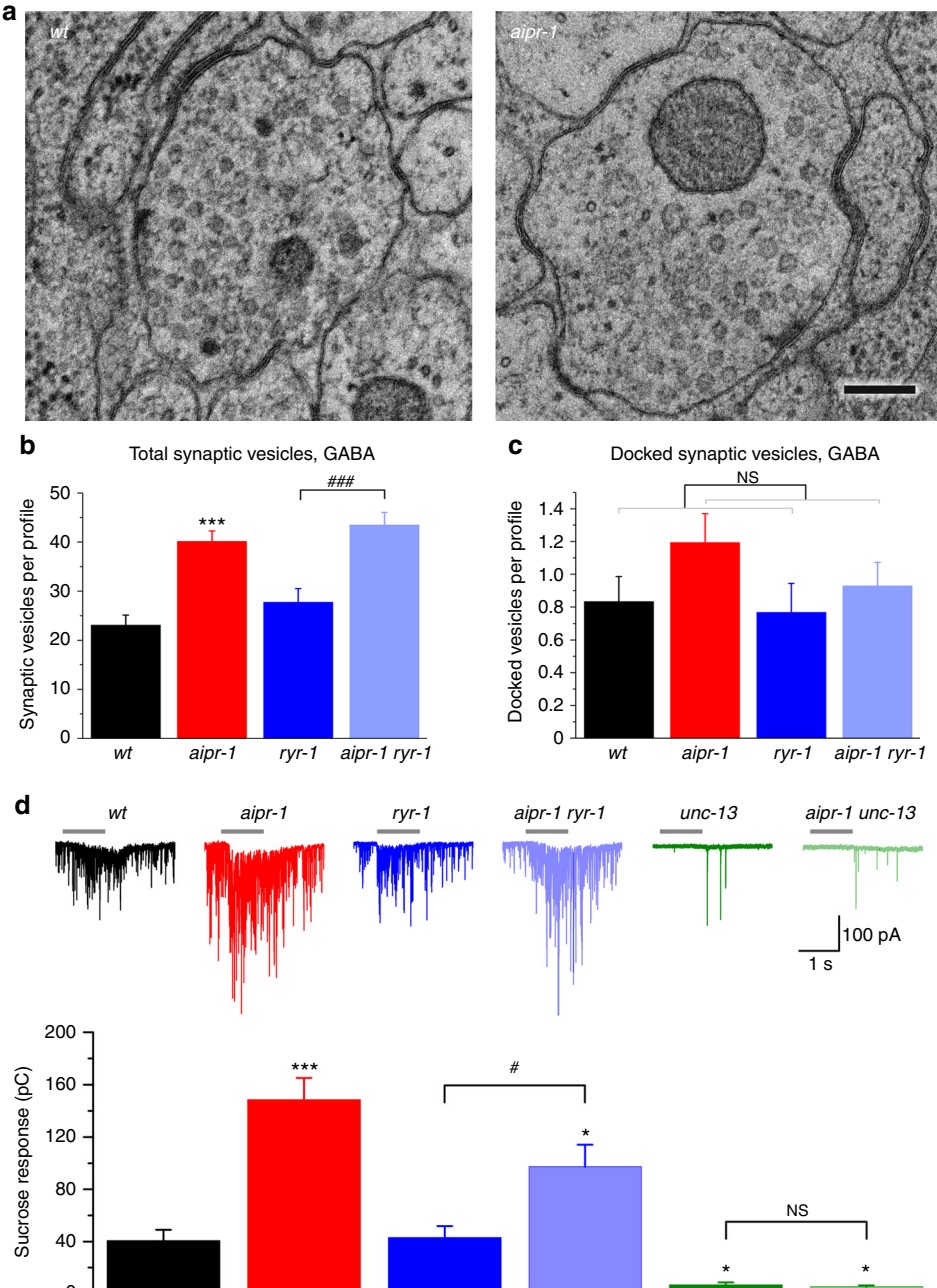

**Fig. 8** Effects of AIPR-1 deficiency on synaptic vesicle number and the size of the readily releasable pool. **a** Sample micrographs of synapses from wild-type and *aipr-1(zw86)* animals. **b**, **c** *airp-1(zw86)* increases the numbers of synaptic vesicles **b** and docked vesicles **c**. Increases were observed regardless of the *ryr-1* genotype. Vesicle numbers compared at GABA synapses among *wt* (n = 66 synaptic profiles), *aipr-1(zw86)* (n = 31), *ryr-1(e540)* (n = 30), and *aipr-1 (zw86) ryr-1(e540)* (n = 56). **d** *aipr-1(zw86)* augmented sucrose-evoked postsynaptic currents at the neuromuscular junction regardless of the presence of *ryr-1* mutation. The sample size was *wt* (n = 7), *aipr-1(zw86)* (n = 8), *ryr-1(e540)* (n = 7), *aipr-1(zw86) ryr-1(e540)* (n = 8), *unc-13(s69)* (n = 7), and *aipr-1 (zw86) unc-13(s69)* (n = 7). Data are shown as mean ± SEM. *$p < 0.05$, ***$p < 0.001$ compared with *wt*; #$p < 0.05$, ###$p < 0.001$, ns $p > 0.05$ compared between indicated groups (one-way ANOVA followed by Tukey's *post-hoc* test or Welch's two-tailed *t*-test (**c**) generalized linear model, Poisson family: *airpr-1(zw86)* effect $p = 0.077$, *ryr-1(e540)* effect: NS. Scale bar, 150 nm

The increase in vesicles in *aipr-1* mutants results in a substantial 3.2-fold increase in primed vesicles as assayed by hypertonic sucrose. Although the increase in docked vesicles (1.4-fold) seems modest by comparison, this value represents docking in single profiles of neuromuscular junctions. Accurate counts of docked vesicles per synapse would require full reconstructions of motor neuron contacts along the muscle. Importantly, the increase in vesicles is not caused by increased calcium release from ryanodine receptors: mutation of RYR-1 does not reverse

the increase in vesicles or the increase in the sucrose response observed in *aipr-1* mutant. At this point it is unclear why vesicles accumulate in *aipr-1* mutant. It is possible that vesicle biogenesis is increased or that vesicle turnover is slowed, or that homeostatic mechanisms of the synapses are disrupted.

The increases in neurotransmitter release, by contrast, can be fully attributed to dysregulation of the ryanodine receptor. The disinhibition of the ryanodine receptor in the *aipr-1* knockdown worms can be directly observed as a doubling of calcium bursts at

synapses. Knockdown of the ryanodine receptor completely eliminates the excessive calcium transients caused by knockdown of *aipr-1*, demonstrating that the calcium barrages arise from internal calcium stores. These calcium bursts are the probable cause of the twofold increase in spontaneous mini frequency observed in the *aipr-1* mutant. It is likely to be that the ryanodine

receptor is also hypersensitive to calcium, as electrical stimulation results in a 2.3-fold increase in evoked responses, suggesting that calcium influx through voltage-gated channels is eliciting a larger response from the ryanodine receptors. Finally, these calcium bursts are driving multiquantal release, as reflected in the 2.7-fold increase in mini amplitudes.

### Table 1 List of worm strains

| Strain ID | Genotype | Source |
|---|---|---|
| Wild type | N2 Bristol | CGC |
| CB246 | *unc-64(e246) III* | CGC |
| CB540 | *ryr-1(e540) V* | CGC |
| CB55 | *unc-2(e55) X* | CGC |
| BC168 | *unc-13(s69) I* | CGC |
| EN208 | *unc-29(kr208::tagRFP) I* | Jean-Louis Bessereau |
| ZW029 | *slo-1(md1745) V* | |
| ZW083 | *zwIs101[Pslo-1::slo-1::GFP (wp5)]* | |
| ZW320 | *zwIs129[Pslo-1::slo-1(gf) (wp708); Pmyo-2::YFP (pPD132.102)]* | |
| ZW742 | *aipr-1(zw86); zwIs101[Pslo-1::slo-1::GFP (wp5)]* | |
| ZW743 | *aipr-1(zw86)II; slo-1(md1745) V* | |
| ZW744 | *aipr-1(zw86) II* | |
| ZW745 | *aipr-1(zw86) II; zwIs129[Pslo-1::slo-1(gf) (wp708); Pmyo-2::YFP (wp214)]* | |
| ZW918 | *aipr-1(zw86) II; zwEx195[Pmyo-3::aipr-1 (wp1466), Pmyo-2::YFP (pPD132.102)]* | |
| ZW929 | *aipr-1(zw86) II; unc-64(e246) III* | |
| ZW932 | *aipr-1(zw86) II; unc-2(e55) X* | |
| ZW936 | *aipr-1(zw86) II; ryr-1(e540) V* | |
| ZW939 | *aipr-1(zw86) II; zwIs133[Punc-17::GFP (wp608)]* | |
| ZW940 | *zwIs138[Prab-3::ryr-1::YFPa (wp1593 + wp1601), lin-15( + )]; zwEx204[Prab-3::aipr-1::YFPc (wp1500), Pmyo-2:: mStrawberry (wp1613)]* | |
| ZW941 | *zwIs138[Prab-3::ryr-1::YFPa (wp1593 + wp1601), lin-15( + )]; zwEx205[Prab-3::aipr-1(Δ1-192)::YFPc (wp1605), Pmyo-2:: mStrawberry (wp1613)]* | |
| ZW942 | *zwIs138[Prab-3::ryr-1::YFPa (wp1593 + wp1601), lin-15( + )]; zwEx206[Prab-3::aipr-1(Δ193-308)::YFPc (wp1606), Pmyo-2::mStrawberry (wp1613)]* | |
| ZW943 | *zwIs138[Prab-3::ryr-1::YFPa (wp1593 + wp1601), lin-15( + )]; zwEx207[Prab-3::AIPR-1(Δ309-342)::YFPc (wp1607), Pmyo-2::mStrawberry (wp1613)]* | |
| ZW944 | *aipr-1(zw86) II; oxIs215[Punc-47::mRFP]* | |
| ZW948 | *zwEx208[Punc-4::aipr-1 senseRNA (wp1611), Punc-4::aipr-1 antisenseRNA (wp1612), Punc-4::mStrawberry (wp1400), Pmyo-2::mStrawberry (wp1613)]; zwEx196[Punc-17Δ1::GCaMP6f (wp1600), lin-15( + )]* | |
| ZW949 | *zwEx209[Punc-4::ryr-1 senseRNA (wp1609), Punc-4::ryr-1 antisenseRNA (wp1610), Punc-4::aipr-1 senseRNA (wp1611), Punc-4::aipr-1 antisenseRNA (wp1612), Punc-4::mStrawberry (wp1400), Pmyo-2::mStrawberry (wp1613)]; zwEx196[Punc-17Δ1::GCaMP6f (wp1600), lin15( + )]* | |
| ZW950 | *zwEx210[Pmyo-2::mStrawberry (wp1613), Punc-4::mStrawberry (wp1400)]; zwEx196[Punc-17Δ1::GCaMP6f (wp1600), lin-15( + )]* | |
| ZW951 | *zwEx211[Punc-4::ryr-1 senseRNA (wp1609), Punc-4::ryr-1 antisenseRNA (wp1610), Punc-4::mStrawberry (wp1400), Pmyo-2::mStrawberry (wp1613)]; zwEx196[Punc-17Δ1::GCaMP6f (wp1600), lin-15( + )]* | |
| ZW1022 | *zwEx232[Pryr-1b::GFP(wp1481), lin15( + )]; lin-15(n765) X* | |
| ZW1028 | *zwEx215[Prab-3::aipr-1::GFP (wp1465), lin-15( + )]; lin-15(n765) X* | |
| ZW1030 | *aipr-1(zw86) II; unc-13(s69) I* | |
| ZW1032 | *aipr-1(zw86) II; zwEx216[Prab-3::aipr-1 (wp1464), Pmyo-2::YFP (wp214)]* | |
| ZW1033 | *aipr-1(zw86) II; zwEx217[Prab-3::mAIP (wp1486), Pmyo-2::YFP (pPD132.102)]* | |
| ZW1098 | *zwEx233[Pryr-1a::GFP(wp149), lin15( + )]; lin-15(n765) X* | |
| ZW1102 | *aipr-1[zw88(GFP::aipr-1)] II* | |
| ZW1103 | *aipr-1[zw89(aipr-1::GFP)] II* | |
| ZW1106 | *zwEx235[Paipr-1::GFP(wp1747 + wp1750); lin-15( + )];lin-15(n765) X* | |
| ZW1111 | *zwEx237[Pmyo-3::ryr-1 senseRNA (wp1745), Pmyo-3::ryr-1 antisenseRNA (wp1746), Pmyo-3::GFP(wp124)]; aipr-1(zw86) II* | |
| ZW1112 | *zwEx238[Prab-3::mStrawberry::PISY-1(wp1387), Prab-3::GFP::ELKS-1(wp1753), lin-15( + )]; lin-15(n765) X* | |
| ZW1113 | *unc-29(kr208::tagRFP) I; aipr-1(zw86) II* | |
| ZW1119 | *zwEx239[Paipr-1::GFP(wp1747 + wp1750), Punc-47::mStrawberry (wp1401), lin-15( + )];lin-15(n765) X* | |
| ZW1120 | *zwEx240[Paipr-1::GFP(wp1747 + wp1750), Punc-17::mRFP (wp865), lin-15( + )];lin-15(n765) X* | |
| ZW1122 | *zwEx241[Prab-3::bkip-4(Δ1-192)::GFP(wp1755), lin-15( + )]; lin-15(n765) X* | |
| ZW1123 | *zwEx242[Prab-3::bkip-4(Δ193-308)::GFP(wp1756), lin-15( + )]; lin-15(n765) X* | |
| ZW1124 | *zwEx243[Prab-3::bkip-4(Δ 309-342)::GFP(wp1757), lin-15( + )]; lin-15(n765) X* | |
| ZW1125 | *zwEx244[Prab-3::aipr-1::GFP (wp1465, 2 ng/μl), Prab-3::TagRFP::elks-1 (wp1670, 5 ng/μl), lin-15( + )]; lin-15(n765) X* | |
| ZW1127 | *zwEx245[Prab-3::GFP::ELKS-1 (wp1753), lin-15(n765) X* | |
| ZW1128 | *zwEx245[Prab-3::GFP::ELKS-1 (wp1753), lin-15( + )]; aipr-1(zw86) II* | |
| ZW1131 | *aipr-1(zw90)/ + II* | |

Strains from CGC (*Caenorhabditis* Genetics Center) and other labs are indicated for their sources

How might AIP regulate the activity of the ryanodine receptor? AIP belongs to the TPR family of co-chaperones that function with HSP90. In contrast to other heatshock proteins that function largely during protein synthesis or in degradation of unfolded proteins, HSP90 often forms stable interactions with intrinsically unstable proteins, such as many kinases and maintains them in a functional state[51]. HSP90 and its co-chaperone AHA1 protect the CFTR channel from thermal instability in the membrane[52]. In a similar manner, AIP may recruit HSP90 to the ryanodine receptor and stabilize the closed state of the ryanodine receptor. In the absence of AIP, the ryanodine receptor is unstable and spontaneously opens to generate calcium bursts from internal stores at the synapse.

Pituitary enlargement is likely an important factor in the hypersecretion of grown hormone in patients with AIP mutations[53]. Based on the results of this study, we speculate that increased calcium release from the ER due to disinhibition of the ryanodine receptor might also contribute to the acromegaly and gigantism phenotypes observed in these patients, although there are cases in which AIP mutations are not associated with hypersecretion of growth hormone[24]. Ryanodine receptors are expressed in the pituitary gland and mediate the release of growth hormone[54,55] and results from goldfish pituitary cells show that pharmacological activation of ryanodine receptors can enhance growth hormone release[56]. Although we have uncovered a potential role for AIP in release at the C. elegans neuromuscular junction, further work in growth hormone-releasing cells is required to verify that AIP deficiencies can cause increased calcium bursts and growth hormone release.

The regulatory effects of AIPR-1 on RYR-1 are supported by our observations that they colocalize at synapses and are in physical contact. The BiFC assays indicate that AIPR-1 binds to RYR-1 through its TPR domain. Structural studies of AIP demonstrate that the TPR domain and the α-7 helix motif are involved in HSP90 binding and in client protein binding[21]. By contrast, the calstabins lack a TPR domain and thus must interact with ryanodine receptors through their N-terminal region[57,58].

The differences in overall structures and in the domains binding to ryanodine receptors suggest that AIPR-1/AIP and calstabins may regulate ryanodine receptor function through different mechanisms. However, it remains to be determined whether they can act cooperatively in inhibition of calcium release from internal stores under physiological conditions.

The present study establishes AIPR-1/AIP as a novel regulator of neurotransmitter release functioning through ryanodine receptors. Neurotransmitter is released by fusion of synaptic vesicles, whereas hormones are released by fusion of secretory granules. Nevertheless, the fusion machinery share many conserved molecules and features, such as the dependence on calcium and the use of the same set of SNARE proteins[59]. The demonstration that AIP is a negative regulator of calcium bursts at synapses opens the possibility that ryanodine receptor disinhibition in endocrine cells might affect secretion of growth hormone.

## Methods

**C. elegans culture and strains**. All worms were raised on agar plates with a layer of OP50 *Escherichia coli* at 22 °C inside an environmental chamber. Strains used in this study are listed in Table 1.

**Genetic Screen**. A genetic screen was performed with an integrated transgenic strain (ZW320) expressing a hyperactive SLO-1 potassium channel, SLO-1(gf), which was created by mutating a single amino acid residue in the cytoplasmic gate of the S6 helix (E350Q)[28]. Synchronized L4-stage slo-1(gf) worms were treated with the chemical mutagen ethyl methanesulfonate (50 mM) for 4 h at room temperature. The F2 progeny were screened for animals that moved better than the original slo-1(gf) animals. Forty mutants were isolated from screening over 100,000 haploid genomes. Complementation tests indicated that many of the mutants belong to genes previously implicated in SLO-1 function or subcellular localization[28,60,61]. zw86 was among five mutants of genes not identified in previous SLO-1(gf)-based genetic screens. Single-nucleotide polymorphism-based genetic mapping[62] of the zw86 strain placed it in a region between 4,951,007 and 11,827,835 on chromosome II. A total of 15 genes within this interval have mutations in this strain based on whole-genome sequencing data. Through complementation tests and mutant rescuing experiments for the candidate genes, we identified zw86 as an allele of C56C10.10. The behavioral phenotypes of zw86 were rescued by a 5.2 kb genomic DNA fragment amplified from wild-type strain using primers 5′-

---

**Table 2 List of primers**

| Gene | Sequence |
| --- | --- |
| aipr-1 | 5′-AATGGATCCGCCACCATGTCGGTCAGAGCAACTGT-3′ (forward) |
| | 5′-ATTACCGGTTATGGCTGAAACATTTTCGAAT-3′ (reverse) |
| mAIP | 5′- ATAAGATCTCCATGGCGGATCTCATCGCAAG-3′ (forward) |
| | 5′-TGTGCCGGCTCAGTGGGAAAAGATGCCC-3′ (reverse) |
| Paipr-1 | 5′- ATACTGCAGCGAGTGGCCGGATAATCTGA-3′ (forward) |
| | 5′-ATTACCGGTGCTCTGACCGACATTCTGAA-3′ (reverse) |
| Pryr-1a | 5′- GTGTTTTCCACGTCTGCGCCTTGCT-3′ (forward) |
| | 5′- ATATTGGCCATCGTTGTCGATCGTCGATCT-3′ (reverse) |
| Pryr-1b | 5′-TCTAAGATCTTGTCATCGTCAATGA-3′ (forward) |
| | 5′-ATTACCGGTACGCAGGAGAGGCAGACGAT-3′ (reverse) |
| aipr-1 RNAi | 5′-TTTGGTACCGTCGGTCAGAGCAACTGTGGT-3′ (forward) |
| | 5′-TATGCCGGCGCTGCGCACATGTGAGTTG-3′ (reverse) |
| ryr-1 RNAi | 5′-TTTGGTACCGGATGGGAGCAAGTGTTTCG-3′ (forward) |
| | 5′- TATGCCGGCTCCGTTGATTGGTTTCCTTCA-3′ (reverse) |
| Pryr-1a | 5′- GTGTTTTCCACGTCTGCGCCTTGCT-3′ (forward) |
| | 5′- ATATTGGCCATCGTTGTCGATCGTCGATCT-3′ (reverse) |
| act-1 | 5′- GCCCAATCCAAGAGAGGTATCC-3′ (forward) |
| | 5′- TGAGGAGGACTGGGTGCTCT-3′ (reverse) |
| C56C10.9 | 5′- TTCCCACAATCTCGACCCA-3′ (forward) |
| | 5′- CAATTCATCAACGGTTCCAGG-3′ (reverse) |
| dnj-8 | 5′- CGCCAATCCAAGAATTCGTT-3′ (forward) |
| | 5′- TGTCCATGCTCCTTCCAGC-3′ (reverse) |
| C56C10.11 | 5′- TCTTGCCAGTCAACCAGATCC-3′ (forward) |
| | 5′- GGAGGTCGAGAAAGGTGGGT-3′ (reverse) |
| pisy-1 | 5′- TAAGGTACCATGGAACCGACAGCCGCCGA-3′ (forward) |
| | 5′- ATAGCTAGCTCATTGGGCCTTCTGAGCAG-3′ (reverse) |

TTCCGGAGTCATGCTGTCC-3′ (forward) and 5′- GTGCTGAG-TATTCTGGGTGCAA-3′ (reverse). The zw86 mutant was outcrossed five times before analyses.

**Rescue and knockout experiments**. Wild-type aipr-1 cDNA (C56C10.10) was amplified from a Bristol N2 cDNA library and cloned into a worm expression vector containing either Prab-3 for neuron-specific rescue (prab-3::aipr-1, wp1464) or Pmyo-3 for muscle-specific rescue (Pmyo-3::aipr-1, wp1466). A mouse AIP cDNA (GenBank: BC075614.1) was amplified from a mouse brain cDNA library and cloned into the worm expression vector containing Prab-3 (Prab-3::mAIP, wp1486). The above plasmids were injected independently into aipr-1(zw86) mutant along with a Pmyo-2::YFP plasmid (pPD132.102), which served as a transformation marker. Primers for cloning aipr-1 and mAIP cDNAs are listed in Table 2.

The CRISPR/Cas9 approach was used to create aipr-1 knockouts[63]. The guide RNA sequence (5′-GTGGTAAAACGGACAATCAG-3′) targeting the first exon of aipr-1 was inserted into Pu6::unc-119 sgRNA (Addgene plasmid 46169) to replace unc-119 sgRNA. This produced plasmid (wp1543) that was coinjected with Peft-3::Cas9-SV40_NLS::tbb-2 (Addgene plasmid 46168) and Pmyo-2::mStrawberry (wp1613) into wild-type worms. Transgenic animals were singled and a genomic DNA fragment of aipr-1 was amplified from the progeny for sequencing. Primers for amplifying and sequencing aipr-1 were 5′-CCATCTCG GATTCTACGCCA-3′ (forward) and 5′-ACAAATCGAAGAGGTCGTGGAT-3′ (reverse).

**GFP tagging of AIPR-1 using Crispr/Cas9**. We fused GFP to the N terminus and C terminus of AIPR-1, respectively, using the Crispr/Cas9 approach. The guide RNA sequences used for the N and C terminus insertions were 5′-ATGTCGGT-CAGAGCAACTG-3′ and 5′- GAACGGAGAGCCGAAAAGA-3′, respectively. The plasmids and the transgenic worms were generated as described above. GFP insertion strains were first screened by PCR followed by sequencing to confirm correct insertions.

**Analysis of expression pattern and subcellular localization**. The expression pattern of aipr-1 was assessed using an in vivo homologous recombination approach. Specifically, a 0.5 kb genomic DNA fragment immediately upstream of aipr-1 initiation site was fused to GFP and the resultant plasmid (wp1747) was linearized and co-injected with a linearized fosmid (WRM065bd01), which contains the entire operon. Recombination was confirmed by PCR. To determine whether aipr-1 is expressed in acetylcholine or GABA neurons, the plasmids Punc-17::mRFP (wp865) and Punc-47::mStrawberry (wp1401) were added to the above injection mixture, respectively. The expression pattern of ryr-1 was assessed by expressing GFP under the control of two alternative ryr-1 promoters (Pryr-1a and Pryr-1b). Primers for cloning Pryr-1a (2.5 kb) and Pryr-1b (4.8 kb) are listed in Table 2. Subcellular localization of AIPR-1 was determined by fusing GFP to its C terminus and expressing the fusion protein under the control of Prab-3 (Prab-3:: AIPR-1::GFP, wp1465). To determine whether AIPR-1 is enriched at presynaptic sites, a Prab-3::TagRFP::elks-1 plasmid (wp1670) was injected into the strain expressing Prab-3::aipr-1::GFP. ELKS-1 is localized at presynaptic sites in C. elegans neurons[64,65]. Worms were immobilized with 10 mM azide in M9 buffer. Images of transgenic worms were taken with a digital CMOS camera (Hamamatsu, C11440-22CU) mounted on a Nikon TE2000-U inverted microscope (Nikon, Tokyo, Japan) equipped with enhanced GFP/fluorescein isothiocyanate (FITC) and mCherry/Texas Red filter sets (49002 and 49008, Chroma Technology Corporation, Rockingham, VT, USA). Primers for cloning Paipr-1 are listed in Table 2.

**Analyses of motor neuron gross morphology and synapse density**. To determine whether motor neurons of the aipr-1(zw86) mutant were grossly abnormal in morphology, two integrated transgenic strains expressing GFP under the control of Punc-17 and mRFP under the control of Punc-47 were first established with wild-type worms. These two transgenes were then independently crossed into aipr-1 (zw86). The numbers of GFP-labeled acetylcholine and mRFP-labeled GABA ventral cord motor neurons, and the overall morphology of motor neuron processes (ventral cord, dorsal cord, and commissures) were compared between wild type and aipr-1(zw86). To determine whether the density of synapses is altered in aipr-1(zw86), whole-mount wild-type and aipr-1(zw86) were immunostained with a primary antibody for RIM[64] and an Alexa Fluor 594-conjugated secondary antibody (Molecular Probes, Eugene, OR, USA). RIM-immunoreactive puncta in a randomly selected segment (60 μm) of the dorsal cord of each worm was manually counted. Worms were visualized and imaged as described above. The exposure time was identical between wild type and aipr-1(zw86) in photographing.

**BiFC assay**. BiFC assays were performed by coexpressing RYR-1 and AIPR-1 tagged with the N- and C-terminal portions of YFP (YFPa and YFPc), respectively, in neurons. As the ryr-1 gene is rather large, the genomic DNA encompassing the neuronal isoforms of RYR-1 (K11C4.5b and K11C4.5d), which were determined through expressing promoter::GFP transcriptional fusions, were amplified as two separate fragments with 750 bp overlap. The amplified upstream and downstream DNA fragments were independently cloned into two worm expression vectors

containing Prab-3 and the DNA sequence encoding YFPa, respectively, to generate plasmids wp1593 and wp1601. In wp1601, the DNA sequence encoding RYR-1 C-terminal was fused in-frame to that encoding YFPa. These two plasmids were linearized and co-injected into lin-15(n765) to produce full-length ryr-1::YFPa fusion via in vivo homologous recombination. A lin-15 rescue plasmid was also coinjected to serve as a transformation marker. The ryr-1::YFPa transgene was integrated into the genome through γ-irradiation. A representative integrant was outcrossed four times and then injected independently with four different plasmids encoding Prab-3::AIPR-1::YFPc (wp1500), Prab-3::AIPR-1(Δ1-192)::YFPc (wp1605), Prab-3::AIPR-1(Δ193-308)::YFPc (wp1606), and Prab-3::AIPR-1(Δ309-342)::YFPc (wp1607). A Pmyo-2::mStrawberry plasmid was coinjected to serve as a transformation marker. Epifluorescence of the transgenic worms was visualized and imaged as described above.

**Recording of postsynaptic currents**. All electrophysiological experiments were performed with adult hermaphrodites. Spontaneous and evoked postsynaptic currents were recorded from the C. elegans neuromuscular junction as described previously[8,45]. To measure the readily releasable pool of synaptic vesicles, PSCs in response to a hyperosmotic sucrose solution (500 mM) were recorded from a body wall muscle cell. The sucrose solution was pressure-ejected (1 s at 10 Psi) onto the ventral nerve cord near the recorded muscle cell through a glass pipette using a Picospritzer III Microinjection Dispense System (Parker Hannifin Precision Fluidics Division, Hollis, NH, USA). Two different extracellular solutions and two different pipette solutions were used, as specified in figure legends. Extracellular solution I contained (in mM) NaCl 140, KCl 5, CaCl$_2$ 5, MgCl$_2$ 5, dextrose 11, and HEPES 5 (pH 7.2). Extracellular solution II differed from extracellular solution I in that CaCl$_2$ was reduced to 0.5 mM, whereas NaCl was increased to 145 mM. Pipette solution I contained (in mM) KCl 120, KOH 20, Tris 5, CaCl$_2$ 0.25, MgCl$_2$ 4, sucrose 36, EGTA 5, and Na$_2$ATP 4 (pH 7.2). Pipette solution II differed from pipette solution I in that 113.2 mM KCl was substituted by Kgluconate. Except when described otherwise in figure legend, all recordings were made with extracellular solution I and pipette solution I at a holding voltage of −60 mV.

**Ca$^{2+}$ imaging and RNAi**. A codon-optimized GCaMP6f was synthesized and placed under the control of Punc-17Δ1, in which the proximal portion (−2,429 to −83 bp) of Punc-17 was deleted so that its promoter activity is restricted almost exclusively to acetylcholine motor neurons in the ventral nerve cord. A-type motor neuron-specific gene knockdown was achieved by coexpressing two plasmids encoding sense and corresponding antisense RNA fragments of the targeted gene under the control of Punc-4. The primers used for amplifying an aipr-1 and ryr-1 cDNA fragments are listed in Table 2. The above cDNA fragments of aipr-1 and ryr-1 were inserted into the Punc-4 expression vector in both orientations to make plasmids encoding corresponding sense and antisense cDNA fragments. The plasmids for aipr-1 and ryr-1 knockdown were injected into the strain expressing Punc-17Δ1::GCaMP6f (wp1600), both separately and in combination, to create three independent transgenic strains. Punc-4::mStrawberry (wp1400) and Pmyo-2:: mStrawberry (wp1613) were coinjected to label A-type motor neurons and serve as a transformation marker, respectively. Young adult worms were immobilized on Sylgard-coated coverglasses by applying a tiny drop of Vetbond Tissue Adhesive (3M Company, St Paul, MN, USA) on the dorsal anterior part of the worm and immersed in extracellular solution I. Spontaneous fluorescence changes in motor neurons of a ventral cord segment anterior to the vulva were imaged at 16 frames per second for 2 min using an electron-multiplying charge-coupled device (CCD) camera (iXonEM + 885, Andor Technology, Belfast, Northern Ireland), a FITC/Texas Red filter set (59222, Chroma Technology Corporation), a light source (Lambda XL, Sutter Instrument, Novato, CA, USA), and NIS-Elements software (Nikon).

**Thrashing assay**. A single young adult hermaphrodite was transferred to an unseeded NGM plate and overspread with ~ 60 μl of M9 buffer. After a 1 min recovery period from the transfer, snapshots of the worm were taken at 15 frames per second for 1 min using a VGA FireWire camera (XCD-V60, Sony, Tokyo, Japan) mounted on a stereomicroscope (SMZ800, Nikon). The worm was automatically kept near the center of the view field through a motorized microscope stage (OptiScan ES111, Prior Scientific, Rockland, MA, USA). Both the camera and the motorized stage were controlled by an automated worm tracking system running in MATLAB (The MathWorks, Natick, MA, USA). Thrashing was counted manually from playing back the movies at 5 frames per second. A "thrash" was defined as one head bending toward one side of the body axis.

**High-pressure freezing**. Young adult worms were high-pressure frozen with a HPM100 (Leica Microsystems, Buffalo Grove, IL, USA). To mount the worms for freezing, a 3 mm support ring was loaded into a standard 6 mm middle plate. A 3 mm sapphire disc (Technotrade International, Manchester, NH, USA) was placed in the support ring, and on the disc was placed a copper 1 × 2 mm transmission electron microscope (TEM) slot grid (Ted Pella, Redding, CA, USA), wetted with hexadecene oil. Using a paintbrush (number 00), ~ 10 worms were transferred to the slot, using the OP50 bacteria as a space filler. A 6 mm B-type aluminum specimen carrier (Ted Pella), flat side down, capped off the assembly. After high-

pressure freezing, the sample (typically still associated with the grid and sapphire) was liberated from the other pieces under −90 °C anhydrous acetone in an automatic freeze substitution chamber (AFS2, Leica Microsystems). The specimen was then transferred to a cryovial containing 1% osmium tetroxide, 1% glutaraldehyde, and 1% water in acetone (Electron Microscopy Sciences, Hatfield, PA, USA), already in the AFS chamber.

**Freeze substation and sample processing.** Freeze substitution was performed with the following program: −90 °C for at least 5 h, + 5 °C/h to −20 °C, −20 °C for 14 h, + 10 °C/h to room temperature. During the program, the vials were agitated at least twice a day, promoting efficient diffusion of the fixatives into the tissues. Once at room temperature, the samples were removed from the AFS chamber and washed with anhydrous acetone six times over 90 min, stained *en bloc* with 0.1% uranyl acetate (Polysciences, Warrington, PA, USA), and washed again with anhydrous acetone six times over 90 min. Next, the worms were nutated while infiltrating with Epon/Araldite resin (Ted Pella) in a stepwise manner: 30% 3 h, 70% 4 h, 90% overnight at 4 °C, and finally three exchanges of 100%, 2 h each. Worms were dissociated from the bacteria and distributed about the bottom of an embedding capsule cap (Ted Pella). Resin was cured for 48 h at 60 °C. Worms were mounted on base blocks and trimmed in cross-section to the anterior reflex of their gonad. Ultrathin (33 nm) serial sections were cut using an EM UC7 (Leica Microsystems) and collected on pioloform-coated slot grids. Post staining was performed with 2.5% uranyl acetate in 70% methanol for 4 min.

**Microscopy.** The ventral nerve cord of the animal was imaged at 120 keV on a JEOL JEM-1400 Plus transmission electron microscope equipped with an Orius CCD camera (Gatan, Pleasanton, CA, USA). Images were collected from typically ~ 300 serial sections from each worm ($n = 2$ worms per genotype). Imaging and analysis were performed blind to genotype.

**Morphometry.** Morphometry was performed with the aid of custom ImageJ (NIH, Bethesda, MD, USA) and MATLAB (MathWorks) scripts (Shigeki Watanabe, M. Wayne Davis, Edward J. Hujber, and Erik M. Jorgensen). Synapses were defined as profiles containing a dense projection plus one adjacent profile on either side. Synaptic vesicles were classified as docked when in direct contact with the plasma membrane.

**Data analyses.** Amplitudes of evoked currents were quantified using Clampfit (version 10, Molecular Devices, Sunnyvale, CA, USA), whereas the frequency and mean amplitude of spontaneous minis using MiniAnalysis (Synaptosoft, Decatur, GA, USA), as described previously[8,45]. The current integral over 1.5 s upon the application of the hyperosmotic sucrose solution was quantified using Clampfit for statistical comparison. Calcium transients of A-type acetylcholine motor neurons were identified based on mStrawberry marker expression and synchronous activities[66]. Transients were quantified by plotting out $F/F_0$ of each A-type motor neuron in the imaging field (typically three or four neurons) over the recording period and measuring the frequency, amplitude, and duration of peaks. Calcium transient data of all the imaged neurons in each prep were averaged to represent one sample for statistical analyses. The detection threshold was set at 0.05 above the baseline, and at least 3 s in duration. When quantifying the number of calcium transients that merged together, a drop of $F/F_0$ amplitude by at least 25% was used as the criteria to indicate that subsequent $F/F_0$ belonged to another calcium transient. Data graphing and statistical analyses were performed with OriginPro 2015 (OriginLab Corporation, Northampton, MA, USA). Data are shown as mean ± SE. Either analysis of variance with Tukey's *post-hoc* test, *t*-test, or a generalized linear model was used for statistical comparisons as specified in figure legends. $p < 0.05$ is considered to be statistically significant. The sample size ($n$) equals to the number of cells, worms, or synaptic profiles analyzed.

**Data availability.** The data that support the findings of this study, including morphometry scripts, are available from the corresponding author upon reasonable request.

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

## Acknowledgments

We thank the *Caenorhabditis* Genetics Center for mutant strains, Jean-Louis Bessereau for EN208 strain, and Fred Adler for advice on statistics. Supported by National Institute of Health grants R01-MH085927 (Z.-W.W.), R01-GM113004 (B.C.), and R01-NS034307 (E.M.J.). E. M. J. is an Investigator of the Howard Hughes Medical Institute (HHMI).

## Author contributions

B.C. did molecular, cellular, and behavioral experiments. P.L. did electrophysiological experiments. E.J.H. did electron microscopy. Y.L. did mutant screen and mapping. B.C. and Z.-W.W. designed experiments. B.C., Z.-W.W. and E.M.J. wrote the manuscript.

## Additional information

**Competing interests:** The authors have no competing financial interests.

