## [Peer Review File · Nature Communications]

Reviewers' comments:

Reviewer #1 (Remarks to the Author):

This paper suggest an entirely new mechanism of effect for the co-chaperone protein AIP. The authors identified that AIP can inhibit calcium release by the ryanodine receptor in *C. elegans* and have a set of elegant studies to prove their point.

One of the major issues is that the authors consider the AIP-related disease as a misregulation of GH-release from somatotrop cells (see first line of Abstract: "AIP ...associated with hypersecretion of growth hormone...") and do not seem to mention in the manuscript and therefore might not appreciate that lack of functional AIP protein primarily causes a tumor, a rather invasive one usually, and the GH or GH & prolactin or sometimes prolactin only excess is a result of that. In addition, there are AIP mutation positive patients with a pituitary tumor without clinical GH excess, although GH staining positive on immunochemistry and very few patients have been described with other pituitary adenoma subtypes (gonadotrophin origin and corticotropic cell origin). If the disease mechanism would purely linked to ryanodine receptor activation, then we would see patients with 'pure' pituitary GH hypersecretion (i.e. without a pituitary tumor) with normal pituitary size and excess GH release. This scenario indeed can be found in some cases of GPR101 duplication-related GH excess or, rarely, to ectopic GHRH secretion-related GH excess, but this has never been described in AIP-related disease. Therefore, while the physiological relevance of the presented data on the inhibitory effect of AIP on ryanodine receptor activity is compelling, the direct clinical relevance the author state relating to AIP mutation positive disease cannot be accepted with the listed arguments.

Authors do not mention any other known function of AIP and how the ryanodine receptor effect would relate to those. Based on the clinical and functional data AIP functions as a tumor suppressor gene and involved in cellular growth and proliferation, link to the ryanodine effect need to be discussed and studied.

The authors did not perform studies in GH-secreting primary cells or cell lines to show the clinical relevance of their finding.

Specific comments

In general, references to the other mechanisms of effects of AIP suggested in other publications is entirely missing.

Line 58 mention that these AIPR-1 mutations are loss of function or not

Line 63 – a leaky ryanodine receptor does not explain the AIP mutation positive patient phenotype, see arguments above

Line 75 it is unclear if this animal was hetero or homozygote for this genotype.

Line 83, define minis

References- Journal name style is inconsistent in the reference list

Reviewer #2 (Remarks to the Author):

The manuscript by Chen et al. describes a very nice study of the *C. elegans* AIRP-1 protein, and for the 1st time, link its function to the ryanodine receptor. There is clinical interest in AIRP-1, due to its disease link to growth hormone hypersecretion. Using the *C. elegans* model, the authors find data to support that AIRP-1 normally acts to limit ryanodine receptor activity. When it is mutated, excessive calcium efflux from the ER via the ryanodine receptor triggers excessive neurotransmitter release at the worm NMJ, providing a potential model for the clinical observations in humans. Even beyond the disease link, the data set is very nice in helping to establish the function of a poorly studied protein, while highlighting how ryanodine receptors are also driving calcium signaling to modify synaptic transmission. I was very impressed with the initial screen to find AIRP-1 mutants – the authors performed a suppressor screen starting with a hyperactive slo mutation that decreases release. AIRP-1 was identified as a suppressor, and the authors then worked back from identifying the locus to showing how it enhances synaptic transmission. The final tie in to the ryanodine receptor biology is nicely supported by the genetics, so I'm quite impressed with the overall data set and think the work will be well received in the field.

In terms of the current manuscript, I'm happy with the data set the authors have provided. One could quibble over the lack of a direct mechanism for how AIRP-1 controls ryanodine receptor function (does it alter aspects of its subcellular localization on the ER, does it prevent other proteins from assembling onto the channel, or does it directly act to normally keep the channel closed?), but I think that mechanism can be a story on its own.

Likewise, the final figure that indicates AIRP-1 has effects on release outside of the ryanodine receptor, through what appears to be alterations in vesicle density at synapses, provides another wrinkle that will need to be worked out in future studies, as its sort of just an observation that is left hanging at the end of the story (and could in fact contribute to why GH secretion is enhanced), but I think it is fine as is with the current strength of the work demonstrating a novel link between AIRP-1, ryanodine receptor hyperactivity, increase intracellular calcium at synapses and excessive neurotransmitter release. Overall a very nice data set.

Reviewer #3 (Remarks to the Author):

In this work Chen et al identified *aipr-1*, which encodes the *C. elegans* ortholog of the aryl hydrocarbon receptor-interacting protein AIP, in a genetic screen for suppression of decreased neurotransmitter release caused by a hyperactive BK channel. The authors investigated the role of AIRP-1 at the neuromuscular junction of *C. elegans* and raised 2 main conclusions:

- AIRP-1 inhibits synaptic transmission at the presynaptic level
- AIRP-1 decreases the Ca²⁺ leak through presynaptic ryanodine receptors.

This study is of broad interest since AIP was shown to be involved in pituitary tumors, and

mutations in the AIP gene are linked to acromegaly and gigantism due to hypersecretion of growth hormone in human patients. The current data provide the first cellular mechanism linking AIP proteins and hypersecretion, which might be conserved in mammals based on experimental data provided in this study.

Overall the manuscript is well written and provides high quality data. However, few points remain to be clarified:

1/ *aipr-1* seems to be embedded in a complex genetic locus since it is the third gene of a predicted operon containing four genes. The entire study is based on the analysis of a single allele, only partially characterized, which is very problematic:

- what is the evidence that two alternative isoforms are produced in *zw86*? RT-PCR analysis of *aipr-1* transcripts in *zw86* should be provided to confirm *in silico* predictions and rule out cryptic or aberrant splicing in this operon.

- is *zw86* affecting other genes in the operon? This question is important because an *aipr-1*-containing fragment was demonstrated to rescue the electrophysiological phenotypes of *zw86* but not the electron microscopy defects. We agree that EM analysis of a rescued strain would represent a substantial amount of work. As an alternative, transcript levels of the upstream and downstream genes in the operon should be measured by RT-PCR in *zw86*.

- similarly, RNAi experiments (fig. 5) are difficult to interpret because it might affect other genes in the operon. This caveat should at least be mentioned.

- is *zw86* causing partial or complete loss of function? is the mutation fully recessive? or is there any evidence for haploinsufficiency in hets, as suggested in human patients?

- generating a null allele by CRISPR would have been extremely informative. The negative results provided in the Material and Methods do not demonstrate that getting a null allele is out of reach. Rather than looking for random deletions, introducing a point mutation by homologous recombination or replace *aipr-1* ORF by a selection marker might be more successful. Until a heterozygous line segregating homozygous loss of function mutants is not generated, it is not possible to state that "AIP is an essential gene in both worms and mammals"

- the structure of the transcriptional reporter constructs is confusing because it basically contains the entire upstream gene of the operon. What is the rationale for picking up this fragment? A proper reporter would require to include potential regulatory sequences upstream to the first gene, either using fosmid recombineering or CRISPR strategies. If these experiments are not performed, it should be clearly stated in the text that the expression pattern based on the current reporter construct might not reflect endogenous *aipr-1* expression.

2/ the electrophysiological phenotype of *aipr-1* mutants is quite peculiar since the occurrence of very large mPSCs is rarely seen in mutants affecting presynaptic release. The current data do not exclude that *aipr-1* mutation causes post-synaptic defects since the currents elicited by ACh and GABA pressure-ejection do activate both synaptic and extrasynaptic receptors and it remains possible that a redistribution of receptors to synaptic domains occurs in *aipr-1* mutants.

- To eliminate secondary developmental defects, we suggest to acutely block RYR-1 with ryanodine, as previously done by this group (Liu et al, 2005), and confirm that it causes the

disappearance of the large mPSCs in zw86.

- the authors propose that large mPSCs in *aipr-1* are caused by multivesicular release based on the phenotype of zw86; *unc-64(e246)* double mutants. This is quite indirect. The authors refer to two previous studies done in rat brain slices. However, they have previously shown that in *C. elegans* muscle large-amplitude minis are not due to multivesicular release (Liu et al., 2005, *J Neuro*). What is going on in *aipr-1* mutants? mPSCs kinetics should be reanalyzed to provide additional evidence for multivesicular release, such as increased rise time of large-amplitude mPSCs, otherwise the evidence for such a mechanism is rather weak at the moment.

3/ One important point of this study is the evidence that UNC-68 and AIPR-1 might interact in vivo. The BiFC assay data (current Supplemental Fig5) thus constitute a key result and should be shown in the main figures. It would be nice to have quantitative data to show the reproducibility of these experiments. Moreover, the AIPR-1 Δ TPR::YFPc provides a critical negative control to support specificity of the BiFC assay, but there is no indication that this protein is actually produced and stable in transgenic animals. Western blot experiments are required to compare the relative expression of the different YFPc fusions among the strains. In addition, the authors suggest that AIPR-1 is enriched at synapses based on the co-localization of AIPR-1 and the active zone protein ELKS-1 (Fig3e). I have few concerns with this experiment. First, to which extent is AIPR-1:GFP functional? Data should be provided in the Supplemental. Second, AIPR-1:GFP is likely massively overexpressed, which might explain the diffused phenotype. Generation of single copy transgene (or endogenous locus tagging by homologous recombination) and quantitative analysis is required before stating that AIPR-1 is enriched at synapses.

4/ The interpretation of increased calcium burst frequency in the soma of motoneurons after *aipr-1* RNAi is difficult. The disappearance of these bursts in *ryr-1* mutants strongly suggests that calcium is released from the ER. Yet, increased frequency after *aipr-1* RNAi might be equally be caused by increased excitability of the motoneurons generating increased CICR. This should at least be discussed.

In addition, the results of the genetic interaction between *unc-2* and *aipr-1* is a bit over interpreted. The fact that "the *unc-2 aipr-1* double mutant exhibits increased neurotransmission compared to *unc-2* alone (Fig. 4a,b)" does not necessarily "demonstrates that AIPR-1 acts in a parallel pathway to the N-type calcium". Those are epistasis experiments performed with alleles that might be non null mutations. These results suggest that at least part of the AIPR-1 function does not imply UNC-2.

Minor points:

- since *unc-2*, *aipr-1*, *unc-64* and *unc-68* are all on different chromosomes, a semicolon should separate mutations in the double mutant genotype according to the *C. elegans* nomenclature. Please check the text and the figures.
- Line 45: Umanskaya A et al is cited as an article whereas it is an abstract of a poster at the Biophysical Society 58th meeting.
- Line 48 : « an α -7 helix » is misleading. What does it mean? It is simply the 7th α -helix of the protein. Please clarify and remove "7" if not necessary.

- Line 90-91: Supplementary Fig 1 is not the right one. Currently it is the Supplementary Fig 3.
- Line 121: Supplementary Fig 3 is not the right one. Currently it is the Supplementary Fig 1.
- Line 152: the thrashing phenotype of *aipr-1* mutant is striking. Just by curiosity, is there any equivalent effect on crawling?
- Line 185-188: there is a mixing up here when referring to the figures Fig6, Supplementary Fig6 and Supplementary Fig7. Please check it carefully and edit this part.
- Line 204: remove the "s" from "decreases" or the one from "demonstrates".
- Line 245: could the authors summarize briefly what is known about the role of RyR in growth hormone secretion in the pituitary gland?
- Line 296: please add a sentence to specify how the worms were immobilized for imaging.
- Line 416: how many A-type motor neurons were observed in the field and averaged by worm? How is a peak defined? Whether it seems obvious for wt, *ryr-1* RNAi and the double RNAi, it is less clear in *aipr-1* mutant. Was a minimum $\Delta F/F$ value fixed?
- Figures: some spaces between the numbers and pA are missing in Fig2b, Fig4a, Fig 5b, FigS3a,b.
- Fig3b-e: could the authors indicate precisely which regions they imaged?
- Fig4b,c: the first letter of *aipr-1* is not italicized properly in the double mutant.
- Fig5b: Could the authors also indicate the result of the statistical post hoc test between *aipr-1* and *aipr-1 unc-64*?

Reviewer #4 (Remarks to the Author):

This is an important study by Chen et al. that describes the function of the newly identified gene, *aipr-1*, in the *C.elegans* NMJ. *aipr-1* is related to human AIP, a gene involved in growth hormone (GH) hypersecretion. Through a combination of methods (electrophysiology, calcium imaging, genetics, electron microscopy, etc), the authors show that *aipr-1* limits neurotransmitter release, a finding highly relevant to the GH hypersecretion defect seen in humans with AIP mutations. At the mechanistic level, this study proposes a dual role for AIPR-1 in inhibiting ryanodine receptor (*ryr-1*) function and synaptic vesicle biogenesis. Although the electrophysiological and EM data are impressive and convincing, several concerns still remain regarding the subcellular localization of AIPR-1, its physical association with RYR-1, and the argument that hypersecretion in *aipr-1* is not caused by a developmental defect.

Major Concerns:

1. The abstract states that AIPR-1 is physically associated with the ryanodine receptor (RyR-1) at synapses. In addition, there is an entire paragraph in Discussion on this. However, the authors need to show convincing evidence if they insist on this claim. The BiFC assay shows no signal at synapses. Since RyR-1 is localized at the ER, does the ER extend up to the pre-synaptic zone? A marker for the ER could resolve whether AIPR-1 and RyR-1 are physically associated at the ER close to the presynaptic terminal. This is

particularly important because there is no general agreement in the literature that RyRs are localized to presynaptic terminals.

2. The authors use an AIPR-1 translational fusion in the context of transgenic (multi-copy array) animals. They mention that this rescues the *aipr-1* behavioral phenotype but no data are shown. It is critically important to show if the *aipr-1* behavioral phenotype is partially or completely restored because this will be informative for whether this *aipr-1* translational reporter is a faithful reporter to monitor AIPR-1 subcellular localization. An antibody or tagging the endogenous *aipr-1* with *gfp* or *tagrfp* using CRISPR (single-copy tagging) could really strengthen the conclusions regarding AIPR-1 subcellular localization. In addition, more presynaptic markers (RIM, SYD-1, SNB-1) are needed to establish AIPR-1 localization at the pre-synapse. The image with ELKS-1 is over-exposed, and therefore not convincing.

3. The characterization of motor neuron synapse development in *aipr-1* mutants described in Suppl Fig 2 needs to be more comprehensive. The authors need to use additional markers of pre-synapse (SYD-1, SNB-1) and post-synapse (UNC-29 and UNC-49). This is particularly important in light of the fact that *aipr-1* has an effect on synaptic vesicle biogenesis. To rule out the possibility of a developmental defect that could account for the increased neurotransmitter release in *aipr-1* mutants, the authors could apply pharmacological agonists and/or antagonists of RyR-1 only at adult stages.

4. More details on the transcriptional fusions for *ryr-1a* and *ryr-1b* are needed. Based on the well-established expression pattern of *ryr-1* in muscle, more details are needed for the two promoters used, especially because there is another gene really close and upstream of the *ryr-1* locus. On a similar note, it will be comforting to see what happens to amplitude and minis after muscle-specific RNAi for *ryr-1* in the *aipr-1* mutant. The text (Results, line 137) is a bit inaccurate because it states that *C.elegans ryr-1* is only in the nervous system.

Minor Points:

1. At the beginning of the Results, it must be clearly stated that this is NMJ recordings, so that the reader gets familiar with the system at the very beginning of the text.
2. Suppl Fig 1 mixed up with Suppl Fig 3. Slo-1 localization in DNC seems affected. Better quantification is needed.
3. Line 104. Mouse AIP rescued AIPR-1 expression. No data for this claim is provided.
4. How many times were the *aipr-1(zw86)* mutants outcrossed before the comparison with *slo-1(gf)* mutants was performed?
5. Line 188 and Fig 6b: the authors do not provide an explanation as to why the *aipr-1;ryr-1* double mutants has a remarkable increase in synaptic vesicles compared to single mutants.
6. Line 209: More than one target is also possible.

Point-by-point response to reviewers.

Reviewers' comments have been renumbered for easy reference.

Responses are labeled, and text changes are quoted.

Reviewer #1

Comment 1

This paper suggests an entirely new mechanism of effect for the co-chaperone protein AIP. The authors identified that AIP can inhibit calcium release by the ryanodine receptor in *C. elegans* and have a set of elegant studies to prove their point.

One of the major issues is that the authors consider the AIP-related disease as a misregulation of GH-release from somatotrop cells (see first line of Abstract: "AIP ...associated with hypersecretion of growth hormone...") and do not seem to mention in the manuscript and therefore might not appreciate that lack of functional AIP protein primarily causes a tumor, a rather invasive one usually, and the GH or GH & prolactin or sometimes prolactin only excess is a result of that. In addition, there are AIP mutation positive patients with a pituitary tumor without clinical GH excess, although GH staining positive on immunochemistry and very few patients have been described with other pituitary adenoma subtypes (gonadotrophin origin and corticotropic cell origin). If the disease mechanism would purely linked to ryanodine receptor activation, then we would see patients with 'pure' pituitary GH hypersecretion (i.e. without a pituitary tumor) with normal pituitary size and excess GH release. This scenario indeed can be found in some cases of GPR101 duplication-related GH excess or, rarely, to ectopic GHRH secretion-related GH excess, but this has never been described in AIP-related disease. Therefore, while the physiological relevance of the presented data on the inhibitory effect of AIP on ryanodine receptor activity is compelling, the direct clinical relevance the author state relating to AIP mutation positive disease cannot be accepted with the listed arguments.

Response

The reviewer is quite right. AIP is mainly known as a suppressor of pituitary tumors. It is not clear how disinhibition of the ryanodine receptor could result in tumor generation and we therefore avoided speculative claims about the molecular mechanisms of tumorigenesis. Unfortunately, by focusing on growth hormone secretion, we ignored the role of AIP in cancer and thereby misrepresented the broader functions of the protein to potential readers. We made the two following changes to the manuscript to address this thoughtful comment of the reviewer.

p. 1, Abstract: "Mutations in the *AIP* gene are a major cause of pituitary tumors, which are generally associated with hypersecretion of growth hormone."

p. 2, Introduction: "AIP is mainly known as a suppressor of pituitary tumors²². Affected individuals are heterozygous for the mutant allele; the tumors are usually homozygous due to loss of heterozygosity in the pituitary gland in both humans and mice²²⁻²⁴."

Comment 2

Authors do not mention any other known function of AIP and how the ryanodine receptor effect would relate to those. Based on the clinical and functional data AIP functions as a tumor suppressor gene and involved in cellular growth and proliferation, link to the ryanodine effect need to be discussed and studied.

Response

The broader consensus seems to be that the role of AIP and HSP90 in stabilizing ligand-activated transcription factors is a more likely cause of tumorigenesis although the roles and interactions of AIP might not yet be fully described. By contrast, a role for the ryanodine receptor in hypersecretion is more likely. Three changes were made to address this comment:

(1) The manuscript now describes:

p.2. "AIP is mainly known as a suppressor of pituitary tumors".

(2) We explicitly state that it is unclear how disinhibiting the ryanodine receptor could be tumorigenic.

p.2. "It is unclear why AIP mutations cause pituitary tumors, although the aryl hydrocarbon receptor-cAMP-phosphodiesterase pathway appears to be important²⁵."

(3) We note that hyperplasia alone could account for hypersecretion, and describe how disinhibition of the ryanodine might lead to hypersecretion.

p.7. "How do these results influence our understanding of the role of AIP in human disease? Although the enlargement of the pituitary alone in tumors will lead to hypersecretion of growth hormone, increased calcium release from the ER due to disinhibition of the ryanodine receptor by AIP might also contribute to the acromegaly and gigantism phenotypes observed in patients. Ryanodine receptors are expressed in the pituitary gland and mediate the release of growth hormone^{53,54}. In the disease state, calcium bursts in the somatotrophic cells of the anterior pituitary gland could lead to inappropriate release of growth hormone, which would lead to gigantism and acromegaly. However, the calcium bursts do not lend a ready explanation for the induction of tumors."

Comment 3

The authors did not perform studies in GH-secreting primary cells or cell lines to show the clinical relevance of their finding.

Response

We thank the reviewer for this excellent suggestion. Indeed, the next logical step is to determine whether a deficiency of AIP in GH-secreting primary cells or a pituitary cell line may cause enhanced growth hormone release due to disinhibition of ryanodine receptors. Although we consider this expanded research direction beyond the scope of this study, we are planning on broadening our research program to investigate the potential clinical relevance of our finding using a variety of approaches, such as RNA interference, optogenetic stimulation, biochemistry, electrophysiology, and electron microscopy, and we will seek collaborators embedded in the medical community with access to patient samples.

Specific comments

Comment 4

In general, references to the other mechanisms of effects of AIP suggested in other publications is entirely missing.

Response

Yes, we were a bit focused on the ryanodine receptor. To describe the disease more fully, we added several sentences in the Introduction. To incorporate molecular models that have been put forth for AIP function into our study, we speculate on how AIP and HSP90 might stabilize the ryanodine receptor.

p. 2, "AIP is mainly known as a suppressor of pituitary tumors²². Affected individuals are heterozygous for the mutant allele; the tumors are usually homozygous due to loss of heterozygosity in the pituitary gland in both humans and mice²²⁻²⁴. It is unclear why AIP mutations cause pituitary tumors, although the aryl hydrocarbon receptor-cAMP-phosphodiesterase pathway appears to be important²⁵. Mutations of the *AIP* gene are also associated with acromegaly and gigantism due to hypersecretion of growth hormone^{22, 26, 27}."

p. 7, "How might AIP regulate the activity of the ryanodine receptor? AIP belongs to the TPR family of co-chaperones that function with HSP90. In contrast to other heatshock proteins that function largely during protein synthesis or in degradation of unfolded proteins, HSP90 often forms stable interactions with intrinsically unstable proteins, such as many kinases, and maintains them in a functional state⁵¹. HSP90 and its co-chaperone AHA1 protect the CFTR channel from thermal instability in the membrane⁵². In a similar manner, AIP may recruit HSP90 to the ryanodine receptor and stabilize the closed state of the ryanodine receptor. In the absence of AIP, the ryanodine receptor is unstable and spontaneously opens to generate calcium bursts from internal stores at the synapse."

Comment 5

Line 58 mention that these *AIPR-1* mutations are loss of function or not.

Response

The *aipr-1(zw86)* is a recessive hypomorphic mutation, which is described in the revised manuscript. To demonstrate that the mutation is recessive, we compared postsynaptic currents at the neuromuscular junction between the wild type and heterozygous *aipr-1(zw86)/+*. Spontaneous and evoked postsynaptic currents were indistinguishable between the wild type and the heterozygote (Fig. 2a, b). To demonstrate that *zw86* is a hypomorph we generated null mutations using CRISPR. Null mutants arrested at early larval stages (Fig. 2c), demonstrating that *zw86* is likely to be a hypomorphic (reduction-of-function) mutant.

P. 3: "Both evoked responses and minis were normal in heterozygous *aipr-1(zw86)*, suggesting that the mutation is fully recessive (Fig. 2a, b)."

p. 3: "A deletion allele of *aipr-1* generated by CRISPR/Cas9 caused animals to arrest at early larval stages (Fig. 2c), which suggest that *aipr-1(zw86)* is a hypomorph."

Comment 6

Line 63 – a leaky ryanodine receptor does not explain the AIP mutation positive patient phenotype, see arguments above.

Response

Although mutations of AIP are strongly associated with pituitary tumors and hypersecretion of growth hormone, clinical presentations of patients with AIP mutations are variable. We agree with the reviewer that the absence of growth hormone hypersecretion in some patients with AIP mutations is inconsistent with a leaky ryanodine receptor. Because AIP may play diverse biological roles through its interactions with different proteins, the clinical complexity might result from differences in the specific AIP mutations or other factors such as narrow tissue-specific loss of heterozygosity. Our observations that *ryr-1* knockout did not occlude the effects of *aipr-1* mutation on the size of the readily releasable pool of synaptic vesicles and the number of synaptic vesicles also suggest that ryanodine receptors are not the only molecular target through which AIPR-1/AIP play biological roles. In any case, we more conservatively propose a contribution by ryanodine receptor destabilization to growth hormone hypersecretion.

p. 7, “How do these results influence our understanding of the role of AIP in human disease? Although the enlargement of the pituitary alone in tumors will lead to hypersecretion of growth hormone, increased calcium release from the ER due to disinhibition of the ryanodine receptor by AIP might also contribute to the acromegaly and gigantism phenotypes observed in patients.”

Comment 7

Line 75 it is unclear if this animal was hetero or homozygote for this genotype.

Response

The *slo-1(gf) aipr-1(zw86)* strain mentioned in line 75 of the original manuscript is homozygous for both mutations. In *C. elegans* literature, homozygous mutants are described without a specific indication – it is assumed. Studies of heterozygous animals are always explicitly stated. For example,

p. 3. “Both evoked responses and minis were normal in heterozygous *aipr-1(zw86)*, suggesting that the mutation is fully recessive (Fig. 2a, b).”

Comment 8

Line 83, define minis

Response

The word “minis” stands for “spontaneous miniature currents”. These currents generally arise from the release of neurotransmitter from a single synaptic vesicle, caused by the spontaneous fusion of synaptic vesicles likely docked near calcium channels. We now define it both at the first use in the Results as well as in figure Legends.

p. 3. “The frequency and mean amplitude of spontaneous miniature postsynaptic currents (‘minis’) ...”

Comment 9

References- Journal name style is inconsistent in the reference list.

Response

We thank the reviewer for the close read. The journal name of one reference (Journal of Endocrinology) was not properly abbreviated during reference formatting. This error has been corrected.

Reviewer #2

Comments

The manuscript by Chen et al. describes a very nice study of the *C. elegans* AIRP-1 protein, and for the 1st time, link its function to the ryanodine receptor. There is clinical interest in AIRP-1, due to its disease link to growth hormone hypersecretion. Using the *C. elegans* model, the authors find data to support that AIRP-1 normally acts to limit ryanodine receptor activity. When it is mutated, excessive calcium efflux from the ER via the ryanodine receptor triggers excessive neurotransmitter release at the worm NMJ, providing a potential model for the clinical observations in humans. Even beyond the disease link, the data set is very nice in helping to establish the function of a poorly studied protein, while highlighting how ryanodine receptors are also driving calcium signaling to modify synaptic transmission. I was very impressed with the initial screen to find AIRP-1 mutants – the authors performed a suppressor screen starting with a hyperactive slo mutation that decreases release. AIRP-1 was identified as a suppressor, and the authors then worked back from identifying the locus to showing how it enhances synaptic transmission. The final tie in to the ryanodine receptor biology is nicely supported by the genetics, so I'm quite impressed with the overall data set and think the work will be well received in the field.

In terms of the current manuscript, I'm happy with the data set the authors have provided.

One could quibble over the lack of a direct mechanism for how AIRP-1 controls ryanodine receptor function (does it alter aspects of its subcellular localization on the ER, does it prevent other proteins from assembling onto the channel, or does it directly act to normally keep the channel closed?), but I think that mechanism can be a story on its own.

Likewise, the final figure that indicates AIRP-1 has effects on release outside of the ryanodine receptor, through what appears to be alterations in vesicle density at synapses, provides another wrinkle that will need to be worked out in future studies, as its sort of just an observation that is left hanging at the end of the story (and could in fact contribute to why GH secretion is enhanced), but I think it is fine as is with the current strength of the work demonstrating a novel link between AIRP-1, ryanodine receptor hyperactivity, increase intracellular calcium at synapses and excessive neurotransmitter release. Overall a very nice data set.

Response

We very much appreciate the positive response of this reviewer. We agree that further studies are needed to fully understand how AIRP-1 regulates the ryanodine receptor and how it regulates synaptic vesicle number independent of the ryanodine receptor. We plan to perform mass spectrometry and a genetic screen for suppressors of the behavioral phenotypes of *aipr-1(w86)* to identify potential AIRP-1-interacting proteins, which should help address the important points raised by this reviewer.

Reviewer #3

Comment 1

This study is of broad interest since AIP was shown to be involved in pituitary tumors, and mutations in the AIP gene are linked to acromegaly and gigantism due to hypersecretion of growth hormone in human patients. The current data provide the first cellular mechanism linking AIP proteins and hypersecretion, which might be conserved in mammals based on experimental data provided in this study. Overall the manuscript is well written and provides high quality data.

Response

We thank the reviewer for the positive appraisal of our manuscript.

Comment 2

aipr-1 seems to be embedded in a complex genetic locus since it is the third gene of a predicted operon containing four genes. The entire study is based on the analysis of a single allele, only partially characterized, which is very problematic.

Response

There are two aspects to this question: the nature of the operon and the reliability of a single allele. The operon issue is addressed below in the responses to Comments 3 through 5.

As suggested by the reviewer, we have isolated another allele. We were unable to isolate additional viable alleles by *slo-1* suppression. The *zw86* allele was the only *aipr-1* mutant isolated in a screen of over 100,000 haploid genomes. This is likely because the behavioral phenotypes of hypomorphic *aipr-1* mutants in the *slo-1(gf)* genetic background are subtle and difficult to spot, we are blinded by suppressors that are specifically required for SLO-1 function, which are profoundly obvious in locomotion phenotypes when screening mutants.

In response, we performed another round of CRISPR mutagenesis, and isolated a null allele in *aipr-1*. Animals homozygous for null alleles arrest at early larval stages as described in the manuscript (Fig. 2c). Although the activity of neurons cannot be tested in this strain, we are confident of the correct gene assignment using standard genetic criteria: 1) *zw86* was mapped to a narrow interval (between 4,951,007 and 11,827,835 on chromosome II), which includes only six identified genes (*egl-44*, *tag-184*, *C56C10.10*, *agr-1*, *cpna-2*, and *D2085.5*); 2) both behavioral and synaptic phenotypes of the *zw86* mutant can be rescued by specifically expressing the *aipr-1* transcript; 3) qPCR of the neighboring transcripts indicated that these genes are not affected by the *zw86* lesion; and 4) RNAi knockdown of the *aipr-1* transcript resulted in similar defects as *zw86*.

p. 3. "We generated a deletion allele of *aipr-1* using CRISPR/Cas9; *zw90* homozygous animals arrest at early larval stages (Fig. 2c), which suggest that *aipr-1(zw86)* is a hypomorph. Similarly, knockouts of the *AIP* gene in mouse are embryonic lethal³²."

Comment 3

-what is the evidence that two alternative isoforms are produced in *zw86*? RT-PCR analysis of *aipr-1* transcripts in *zw86* should be provided to confirm in silico predictions and rule out cryptic or aberrant splicing in this operon.

Response

We thank the reviewer for pointing out this oversight. Indeed, the two alternative isoforms of *aipr-1* in the mutant were identified by RT-PCR, which is now described.

p. 3. "Sequencing of *aipr-1(zw86)* cDNA revealed that the mutant does not make wild-type AIPR-1 but may produce two alternative isoforms with frame shifts (Fig. 1b)."

Comment 4

-is *zw86* affecting other genes in the operon? This question is important because an *aipr-1*-containing fragment was demonstrated to rescue the electrophysiological phenotypes of *zw86* but not the electron microscopy defects. We agree that EM analysis of a rescued strain would represent a substantial

amount of work. As an alternative, transcript levels of the upstream and downstream genes in the operon should be measured by RT-PCR in *zw86*.

Response

To address this comment, we analyzed the levels of transcripts for the other three genes in the operon by RT-PCR. We found that their levels are comparable between wild type and the *aipr-1* mutant (Supplementary Fig. 1), suggesting that the *aipr-1* mutation does not affect the other genes in the operon, and that the synaptic and behavioral phenotypes of the mutant result from AIPR-1 defect. This conclusion is also supported by the observation that neuronal expression of wild-type AIPR-1 alone is adequate for rescuing synaptic phenotypes of the *aipr-1* mutant (Fig. 2).

p. 3. “The *zw86* mutation did not affect mRNA levels of the other three genes in the operon (Supplementary Fig. 1).”

Comment 5

-similarly, RNAi experiments (fig. 5) are difficult to interpret because it might affect other genes in the operon. This caveat should at least be mentioned.

Response

We acknowledge that transitive RNA interference could potentially act on other genes in the operon. Although Andy Fire did not observe transitive effects within an operon, Michel Labouesse detected effects on a pre-mRNA transcript. As suggested by the reviewer, this caveat is mentioned in the revised manuscript.

p. 5. “To knockdown AIPR-1 expression, we performed RNA interference. RNAi is expected to specifically act on *aipr-1* transcripts although we cannot exclude the possibility that RNAi might affect other genes of the operon.”

Comment 6

-is *zw86* causing partial or complete loss of function? is the mutation fully recessive? or is there any evidence for haploinsufficiency in hets, as suggested in human patients?

Response

To address this comment of the reviewer, we compared postsynaptic currents at the neuromuscular junction between the wild type and heterozygous *aipr-1(zw86)*, which was created by crossing with wild-type males. The worms were easily identified because they were similar to wild-type worms, and not small and hyperactive like the homozygous mutant worms. Both spontaneous and evoked postsynaptic currents were indistinguishable between wild type and the heterozygote (Fig. 2a, b), suggesting that the mutation in *zw86* allele is fully recessive. Because putative nulls of *aipr-1* generated by CRISPR arrested at early larval stages (Fig. 2c), *zw86* is likely to be a hypomorphic (reduction-of-function) mutant.

p. 3. “Both evoked responses and minis were normal in heterozygous *aipr-1(zw86)*, suggesting that the mutation is fully recessive (Fig. 2a, b).”

Comment 7

-generating a null allele by CRISPR would have been extremely informative. The negative results provided in the Material and Methods do not demonstrate that getting a null allele is out of reach. Rather than looking for random deletions, introducing a point mutation by homologous recombination

or replace *aipr-1* ORF by a selection marker might be more successful. Until a heterozygous line segregating homozygous loss of function mutants is not generated, it is not possible to state that "AIP is an essential gene in both worms and mammals"

Response

As suggested by the reviewer, we generated *zw90* a putative null mutation in *aipr-1* using Crispr/Cas9. All homozygous mutant worms segregating from the heterozygous parent arrested at early larval stages (Fig. 2c), suggesting that the *aipr-1* gene is essential in worms.

p. 3. "We generated a deletion allele of *aipr-1* using CRISPR/Cas9; *zw90* homozygous animals arrest at early larval stages (Fig. 2c), which suggest that *aipr-1(zw86)* is a hypomorph."

Comment 8

-the structure of the transcriptional reporter constructs is confusing because it basically contains the entire upstream gene of the operon. What is the rationale for picking up this fragment? A proper reporter would require to include potential regulatory sequences upstream to the first gene, either using fosmid recombineering or CRISPR strategies. If these experiments are not performed, it should be clearly stated in the text that the expression pattern based on the current reporter construct might not reflect endogenous *aipr-1* expression.

Response

As the reviewer pointed out, the genomic DNA sequence used in the transcriptional reporter construct included the upstream gene in the operon. This genomic DNA fragment was used in the GFP transcriptional fusion because a genomic DNA fragment encompassing it and the downstream *aipr-1* gene could rescue the behavioral phenotypes of *aipr-1(zw86)* mutant during our genetic mapping. We agree with the reviewer that this is not an appropriate approach for analyzing the expression pattern of *aipr-1* because any upstream sequences important to regulating *aipr-1* expression were missing in the GFP transcriptional construct. As suggested by the reviewer, we re-examined *aipr-1* expression pattern using an *in vivo* homologous recombination approach, in which a linearized fosmid containing the entire operon and a linearized plasmid containing a 0.5-kb fragment upstream of the *aipr-1* initiation site fused to GFP were co-injected into worms. Compared with the previous GFP transcriptional fusion, expression of the new GFP transcriptional fusion was detected in more cell types but fewer neurons in the head (Fig. 3a). Nevertheless, GFP expression in the ventral cord motor neurons was observed with both transcriptional fusions (Fig. 3b, c in both the old and new versions of the manuscript), which explains why the genomic DNA fragment used in genetic mapping could rescue behavioral phenotypes of the *aipr-1* mutant.

p. 4. "To determine the expression pattern of AIPR-1, we expressed a GFP reporter in worms using an *in vivo* homologous recombination approach by co-injecting a plasmid containing 0.5 kb DNA sequence upstream of the *aipr-1* initiation site fused to GFP and a fosmid (WRM065bd01) containing the entire operon. In transgenic worms, GFP was observed in a variety of cell types, including neurons, body-wall muscle cells, amphid sheath cells, spermatheca, and the intestine (Fig. 3a). Both acetylcholine and GABA motor neurons express *aipr-1*, as indicated by the colabeling of these neurons by a red fluorescent protein expressed under the control of cell-specific promoters (Fig. 3b, c)."

Comment 9

The electrophysiological phenotype of *aipr-1* mutants is quite peculiar since the occurrence of very large mPSCs is rarely seen in mutants affecting presynaptic release. The current data do not exclude that *aipr-*

1 mutation causes post-synaptic defects since the currents elicited by ACh and GABA pressure-ejection do activate both synaptic and extrasynaptic receptors and it remains possible that a redistribution of receptors to synaptic domains occurs in *aipr-1* mutants.

- To eliminate secondary developmental defects, we suggest to acutely block RYR-1 with ryanodine, as previously done by this group (Liu et al, 2005), and confirm that it causes the disappearance of the large mPSCs in *zw86*.

Response

As suggested by the reviewer, we examined the acute effect of ryanodine receptor inhibition on postsynaptic currents at the neuromuscular junction of the *aipr-1* mutant. We found that ryanodine (100 μ M) caused great decreases in the amplitudes of both mPSCs and ePSCs (Fig. 4a, b), suggesting that the large mPSCs observed in the mutant did not result from a secondary developmental defect. In addition, we found that there is no obvious difference in the distribution of UNC-29, which is a key subunit of postsynaptic acetylcholine receptors in muscle, between wild type and *aipr-1(zw86)* (Fig. 2e).

p. 5. "Similarly, acute inhibition of ryanodine receptors using 100 μ M ryanodine abolished the increased evoked responses and minis in *aipr-1* mutants (Fig. 4a, b), suggesting that the suppression of defects in *aipr-1* by *ryr-1* mutations is not due to a developmental defect."

p. 4. "the expression of a tagRFP-tagged UNC-29³³, which is a key subunit of muscle acetylcholine receptor³⁴, was similar between the wild type and *aipr-1(zw86)* (Fig. 2e), suggesting that muscle physiology is normal."

Comment 10

- the authors propose that large mPSCs in *aipr-1* are caused by multivesicular release based on the phenotype of *zw86; unc-64(e246)* double mutants. This is quite indirect. The authors refer to two previous studies done in rat brain slices. However, they have previously shown that in *C. elegans* muscle large-amplitude minis are not due to multivesicular release (Liu et al., 2005, J Neuro). What is going on in *aipr-1* mutants? mPSCs kinetics should be reanalyzed to provide additional evidence for multivesicular release, such as increased rise time of large-amplitude mPSCs, otherwise the evidence for such a mechanism is rather weak at the moment.

Response

As suggested by the reviewer, we analyzed the rise time of mPSCs. Our analyses indicate that large mPSCs of *aipr-1(zw86)* have a significantly longer rise time compared with mPSCs of wild type (Fig. 6b), which provides further evidence that they result from multivesicular release.

p. 5. "Furthermore, we analyzed the rise times of minis, which are predicted to increase if large minis are due to multivesicular release^{4, 5}. Our previous study shows that, in wild-type worms, all minis are mono-quantal events with a similar rise time regardless of the amplitude⁶. We divided minis of the *aipr-1* mutant into two categories based on amplitudes (<100 pA and \geq 100 pA), and compared the mean rise times with those of the wild type. We found that the rise time of large minis in the *aipr-1* mutant was significantly longer than that of wild-type minis (Fig. 6b)."

Comment 11

One important point of this study is the evidence that UNC-68 and AIPR-1 might interact in vivo. The BiFC assay data (current Supplemental Fig5) thus constitute a key result and should be shown in the main figures. It would be nice to have quantitative data to show the reproducibility of these experiments. Moreover, the AIPR-1 Δ TPR::YFPc provides a critical negative control to support specificity

of the BiFC assay, but there is no indication that this protein is actually produced and stable in transgenic animals. Western blot experiments are required to compare the relative expression of the different YFPc fusions among the strains.

Response

BiFC was observed in most transgenic worms co-expressing RYR-1::YFPa and YFPc-tagged AIPR-1 of either the full length or with a N- or C-terminal deletion but never with the TPR deletion. These observations suggest that RYR-1 and AIPR-1 physically interact and that the interaction requires the TPR domain. Because BiFC results were of the YES or NO nature, quantification seems to be unnecessary. To address the reviewer's concern over fusion protein expression, we tagged the various fragments of AIPR-1 with GFP and expressed them in wild-type worms. We found that all the GFP fusion proteins were robustly expressed in vivo (Fig. 7c), suggesting that the negative BiFC result with AIPR-1 Δ TPR::YFPc did not result from a reduced protein expression. As suggested, we have now moved this figure to the main text (Fig. 7).

p. 6. "However, YFP fluorescence was not detected when the TPR domain was deleted (Fig. 7b). The lack of BiFC fluorescence with the TPR deletion is unlikely to be due to a lack of protein expression because this deletion did not appreciably alter the expression level of a GFP fusion protein (Fig. 7c)."

Comment 12

In addition, the authors suggest that AIPR-1 is enriched at synapses based on the co-localization of AIPR-1 and the active zone protein ELKS-1 (Fig3e). I have few concerns with this experiment. First, to which extent is AIPR-1::GFP functional? Data should be provided in the Supplemental. Second, AIPR-1::GFP is likely massively overexpressed, which might explain the diffused phenotype. Generation of single copy transgene (or endogenous locus tagging by homologous recombination) and quantitative analysis is required before stating that AIPR-1 is enriched at synapses.

Response

To address this comment of the reviewer, we generated two single-copy GFP insertion strains using the Crispr/Cas9: GFP::AIPR-1 (zw1102) and AIPR-1::GFP (zw1103), in which GFP was fused to the amino- and carboxy-termini, respectively. *aipr-1*(zw86) mutants are hyperactive and *aipr-1* null mutants arrest at early larval stages. By contrast, both single-copy GFP insertion strains displayed normal locomotion, suggesting that the GFP insertion did not disrupt AIPR-1 function. To confirm this conclusion, we performed electrophysiological analyses of the AIPR-1::GFP strain. We found that both evoked responses and minis at the neuromuscular junction were similar to those of the wild type (Supplementary Fig. 6a,b). However, native AIPR-1 expression levels were very low. Although GFP signals were observed in the head region in these two single-copy GFP insertion strains (Supplementary Fig. 6c), it could not be detected along the ventral and dorsal nerve cords by either imaging GFP fluorescence or immunostaining using GFP antibodies, which prevented us from analyzing synaptic localization of the AIPR-1 and GFP fusion proteins. In the original manuscript, AIPR-1::GFP expression along the ventral nerve cord (Fig. 3e) was of a strain in which the Prab-3::AIPR-1::GFP plasmid was injected at 20 ng/ μ l. In this revised manuscript, we injected the Prab-3::AIPR-1::GFP plasmid at a much lower concentration (2 ng/ μ l). AIPR-1::GFP still displays both diffuse and punctate expression along the dorsal nerve cord, and GFP puncta colocalized with the TagRFP::ELKS-1 presynaptic marker (Fig. 3d), supporting the notion that AIPR-1 is enriched at synaptic sites. It appears that technical limitations prevent us from detecting synaptic enrichment of AIPR-1 in the single-copy GFP insertion strains.

p. 4. “To determine the subcellular localization of AIPR-1, we tagged the protein with GFP at either the N- or C-terminus using CRISPR/Cas9. Both strains exhibit wild-type behavior, suggesting that the fusion proteins are functional. Electrophysiological responses were tested in the AIPR-1::GFP strain; evoked responses and minis were normal (Supplementary Fig. 6a, b). However, only very dim GFP fluorescence could be detected in these strains (Supplementary Fig. 6c). The lack of a strong GFP signal in the nerve cords prevented us from analyzing synaptic localization of AIPR-1. To determine AIPR-1 subcellular localization in neurons, we overexpressed AIPR-1::GFP under the control of the panneuronal *rab-3* promoter. The tagged protein was found throughout axons and was enriched at synapses, as determined by colocalization with a presynaptic marker (Fig. 3d).”

Comment 13

The interpretation of increased calcium burst frequency in the soma of motoneurons after *aipr-1* RNAi is difficult. The disappearance of these bursts in *ryr-1* mutants strongly suggests that calcium is released from the ER. Yet, increased frequency after *aipr-1* RNAi might equally be caused by increased excitability of the motoneurons generating increased CICR. This should at least be discussed.

Response

We agree with the reviewer that the ryanodine receptor is not the only target of AIPR-1, since other defects are observed that are independent of RYR-1. However, the increased calcium bursts observed upon AIPR-1 knockdown seem to be fully accounted for by disinhibition of the ryanodine receptor, since in the RYR-1 knockdown strain the additional knockdown of AIPR-1 does not increase burst frequency. Nevertheless, we now explicitly acknowledge multiple targets of AIPR-1 in the Discussion of the revised manuscript.

p. 6. “In addition to the ryanodine receptor, AIP must act on at least one other target that limits synaptic vesicle number at neuromuscular junctions.”

Comment 14

In addition, the results of the genetic interaction between *unc-2* and *aipr-1* is a bit over interpreted. The fact that “the *unc-2 aipr-1* double mutant exhibits increased neurotransmission compared to *unc-2* alone (Fig. 4a,b)” does not necessarily “demonstrates that AIPR-1 acts in a parallel pathway to the N-type calcium”. Those are epistasis experiments performed with alleles that might be non null mutations. These results suggest that at least part of the AIPR-1 function does not imply UNC-2.

Response

As suggested by the reviewer, we have modified our interpretation of the results.

p. 5. “However, in contrast to the *aipr-1 ryr-1* mutant, the *aipr-1 unc-2* double mutant exhibits increased neurotransmission compared to *unc-2* alone (Fig. 4a,b), suggesting that at least part of the AIPR-1 function does not require the N-type calcium channel.”

Minor points:

Comment 15 - since *unc-2*, *aipr-1*, *unc-64* and *unc-68* are all on different chromosomes, a semicolon should separate mutations in the double mutant genotype according to the *C. elegans* nomenclature. Please check the text and the figures.

Response. We appreciate the close read of the genotypes and have made certain that semicolons are present in the strains table. However, this creates problems in sentences and we have not added them

to the text. Moreover, the use of semicolons for non-synteny would wreak real havoc on the clarity of results embedded in the text. For example,; "... the double mutant exhibits reduced evoked and spontaneous currents identical to the *ryr-1* single mutant (evoked: *aipr-1* 5.1 ± 0.5 nA; *ryr-1* 1.1 ± 0.2 nA; *aipr-1 ryr-1* 1.1 ± 0.1 nA; minis: *aipr-1* 73.3 ± 9.9 pA; *ryr-1* 18.2 ± 1.1 pA; *aipr-1 ryr-1* 19.2 ± 0.9 pA; mini frequency: *aipr-1* 107.6 ± 8.6 /sec; *ryr-1* 34.5 ± 5.8 /sec; *aipr-1 ryr-1* 38.9 ± 2.4 /sec) (Fig. 4a,b)." Addition of semicolons between *aipr-1* and *ryr-1* would create a string of unreadable punctuation.

EMJ has thought about this a great deal: Why did fly geneticists select a semicolon to indicate that mutations reside on different chromosomes? The semicolon introduces a full pause and the sentence can occasionally be read as two independent clauses - it interferes with meaning and clarity of the text. I suspect it was a convenience when manuscripts were written on a typewriter; they needed to select a character present among the keys on a typewriter. Ironically, it was Sturtevant himself who demonstrated that synteny did not imply a functional relationship of genes – it doesn't matter in double mutants whether the genes are on the same or different chromosomes. To add to the confusion, semicolons have been adopted to mean different things for different organisms. In mouse nomenclature a semicolon means that the strain has been backcrossed to the inbred strain less than five generations. In any case, the reviewer is quite right – we did not indicate specific chromosomes for the genotypes in the table of strains. We have corrected this oversight.

Comment 16 - Line 45: Umanskaya A et al is cited as an article whereas it is an abstract of a poster at the Biophysical Society 58th meeting.

Response - It is now indicated as an abstract in the reference list.

p. 13. "18.Umanskaya A, Andersson D, Xie W, Reiken S, Marks AR. *C. elegans* as a model to study ryanodine receptor function in aging (ABSTRACT). *Biophys J* **106**, 111a (2014)."

Comment 17 - Line 48 : « an α -7 helix » is misleading. What does it mean? It is simply the 7th α -helix of the protein. Please clarify and remove "7" if not necessary.

Response – The α -7 helix is a structure outside of the TPR motifs and has been found to be an integral part of the TPR domain although it has been named as a separate component from TPR repeats (ref # 19). It is true that it is somewhat unclear to the general reader, but those working on TPR domain proteins will expect to see it there. We now note that it is part of the TPR structure in the introduction and a figure legend:

p. 2. "It contains a prolyl-isomerase-like domain on the amino terminal, and a TPR domain and a helical extension to the TPR domain called the " α -7 helix" on the carboxyl terminal¹⁹ (Fig. 1c)."

Figure 1 Legend: "The prolyl cis-trans isomerase-like domain (Plase-like), tetratricopeptide repeat (TPR) motifs, and the carboxyl terminal α -7 helix (α -Hx) of the TPR repeat structure are underlined."

Comment 18 - Line 90-91: Supplementary Fig 1 is not the right one. Currently it is the Supplementary Fig 3.

Response – We thank the reviewer for pointing out this mistake, which has been corrected.

Comment 19 - Line 121: Supplementary Fig 3 is not the right one. Currently it is the Supplementary Fig 1.

Response - We thank the reviewer for pointing out this mistake, which has also been corrected.

Comment 20 - Line 152: the thrashing phenotype of *aipr-1* mutant is striking. Just by curiosity, is there any equivalent effect on crawling?

Response - We compared the locomotion speed between *aipr-1(zw86)* and wild type using an automated worm-tracking system. As shown in Supplementary Fig. 2, *aipr-1(zw86)* mutant worms move much faster than wild type.

Comment 21- Line 185-188: there is a mixing up here when referring to the figures Fig6, Supplementary Fig6 and Supplementary Fig7. Please check it carefully and edit this part.

Response – now corrected. We have also labeled the panels more clearly.

Comment 22- Line 204: remove the “s” from “decreases” or the one from “demonstrates”.

Response - Corrected.

Comment 23- Line 245: could the authors summarize briefly what is known about the role of RyR in growth hormone secretion in the pituitary gland?

Response – It has been shown that pharmacological activation of ryanodine receptors may enhance growth hormone release from goldfish pituitary cells, which is mentioned in the revised manuscript. p. 7. “This conclusion is supported by results from goldfish pituitary cells showing that pharmacological activation of ryanodine receptors can enhance growth hormone release⁵⁸.”

Comment 24- Line 296: please add a sentence to specify how the worms were immobilized for imaging.

Response – Now included in the Methods.

p. 10, “Young adult worms were immobilized on Sylgard-coated coverglasses by applying a tiny drop of Vetbond™ Tissue Adhesive (3M Company, St. Paul, MN) on the dorsal anterior part of the worm, and immersed in extracellular solution I.”

Comment 25- Line 416: how many A-type motor neurons were observed in the field and averaged by worm? How is a peak defined? Whether it seems obvious for wt, *ryr-1* RNAi and the double RNAi, it is less clear in *aipr-1* mutant. Was a minimum $\Delta F/F$ value fixed?

Response – All A-type motor neurons within the view field (typically 3 or 4) were analyzed. Calcium transient data of all the imaged neurons in each prep were averaged to represent one sample for statistical analyses. The detection threshold was set at 0.05 above the baseline, and at least 3 sec in duration. When quantifying the number of calcium transients that merged together, a drop of F/F_0 amplitude by at least 25% was used as the criteria to indicate that subsequent F/F_0 belonged to another calcium transient. This is now described in the manuscript.

p. 11, “Transients were quantified by plotting out F/F_0 of each A-type motor neuron in the imaging field (typically 3 or 4 neuron) over the recording period and measuring the frequency, amplitude, and duration of peaks. Calcium transient data of all the imaged neurons in each prep were averaged to represent one sample for statistical analyses. The detection threshold was set at 0.05 above the baseline, and at least 3 sec in duration. When quantifying the number of calcium transients that merged together, a drop of F/F_0 amplitude by at least 25% was used as the criteria to indicate that subsequent F/F_0 belonged to another calcium transient.”

Comment26 - Figures: some spaces between the numbers and pA are missing in Fig2b, Fig4a, Fig 5b, FigS3a,b.

Response – Fixed.

Comment 27- Fig3b-e: could the authors indicate precisely which regions they imaged?

Response – Fig. 3 was replaced by a new figure (still the same figure number). The regions imaged are now described in the figure legend.

Figure 3 Legend: “**b, c**, AIPR-1 is expressed in all acetylcholine and GABA motor neurons in the ventral nerve cord. Acetylcholine and GABA motor neurons were labeled by expressing mRFP and mStrawberry under the control of *Punc-17* and *Punc-47*, respectively. Images were taken from a segment anterior (**b**) and posterior (**c**) to the vulva. **d**, AIPR-1::GFP displayed both diffuse and punctate distribution along the dorsal nerve cord, and the GFP puncta colocalized with the presynaptic marker TagRFP::ELKS-1. Images were taken from a dorsal segment anterior to the vulva.”

Comment 28- Fig4b,c: the first letter of *aipr-1* is not italicized properly in the double mutant.

Response – Fixed.

Comment 29- Fig5b: Could the authors also indicate the result of the statistical post hoc test between *aipr-1* and *aipr-1 unc-64*?

Response – They are significantly different, which is indicated in the revised figure (Fig. 6a).

Figure 6 Legend: “Data are shown as mean \pm s.e.m. * $p < 0.05$, *** $p < 0.001$ compared with *wt*; # $p < 0.05$, ### $p < 0.001$, and ns ($p > 0.05$) compared between indicated groups (one-way ANOVA followed by Tukey’s post hoc test).”

Reviewer #4

This is an important study by Chen et al. that describes the function of the newly identified gene, *aipr-1*, in the *C. elegans* NMJ. *aipr-1* is related to human AIP, a gene involved in growth hormone (GH) hypersecretion. Through a combination of methods (electrophysiology, calcium imaging, genetics, electron microscopy, etc), the authors show that *aipr-1* limits neurotransmitter release, a finding highly relevant to the GH hypersecretion defect seen in humans with AIP mutations.

Response: We thank the reviewer for noting the significance of the manuscript.

Major Concerns:

Comment 1

The abstract states that AIPR-1 is physically associated with the ryanodine receptor (RyR-1) at synapses. In addition, there is an entire paragraph in Discussion on this. However, the authors need to show convincing evidence if they insist on this claim. The BiFC assay shows no signal at synapses. Since RyR-1 is localized at the ER, does the ER extend up to the pre-synaptic zone? A marker for the ER could resolve whether AIPR-1 and RyR-1 are physically associated at the ER close to the presynaptic terminal. This is particularly important because there is no general agreement in the literature that RyRs are localized to presynaptic terminals.

Response

As suggested by the reviewer, we co-expressed an ER marker (PISY-1::mStrawberry) and a presynaptic marker (GFP::ELKS-1) in neurons, and analyzed their localization along the dorsal nerve cord. We found that the ER marker was distributed diffusely throughout the nerve cord, including synaptic areas (Supplementary Fig. 8). This apparent proximity of ER to presynaptic sites is consistent with the important roles of ryanodine receptors in neurotransmitter release in *C. elegans* and other species.

To address potentially similar concerns from some readers, we also mentioned in the revised manuscript that the ER “exists in axons and nerve terminals” and cited three selected references. In particular there is a spectacular reconstruction of ER in axons in Wu et al. 2017 (ref # 2).

In regard to ER at synapses in *C. elegans*, we are reconstructing segments of the dorsal nerve cord to characterize the nematode neuromuscular junctions from serial electron micrographs. As shown in the example below (Fig. 1), the ER extends throughout the reconstructed synapse, with a dense projection and synaptic vesicles located close to it.

Fig. 1. Reconstruction of 3 μm from the dorsal nerve cord of a *C. elegans* adult.

Blue - mitochondria
Red - dense projection
Orange - endosome
Yellow tube –ER
Red dots – SVs
Blue dots – LDCVs

In regard to ryanodine receptors at synapses in *C. elegans*, we are localizing calcium channels by super-resolution, and have confirmed their presence at neuromuscular junctions. (Fig. 2)

Fig. 2. RRY-1 is localized at *C. elegans* neuromuscular junctions. Single protein localization by biplane ground-state depletion.

Comment 2

The authors use an AIPR-1 translational fusion in the context of transgenic (multi-copy array) animals. They mention that this rescues the *aipr-1* behavioral phenotype but no data are shown. It is critically important to show if the *aipr-1* behavioral phenotype is partially or completely restored because this will be informative for whether this *aipr-1* translational reporter is a faithful reporter to monitor AIPR-1 subcellular localization. An antibody or tagging the endogenous *aipr-1* with *gfp* or *tagrfp* using CRISPR (single-copy tagging) could really strengthen the conclusions regarding AIPR-1 subcellular localization. In addition, more presynaptic markers (RIM, SYD-1, SNB-1) are needed to establish AIPR-1 localization at the pre-synapse. The image with ELKS-1 is over-exposed, and therefore not convincing.

Response

To generate single copy gene tags we inserted GFP at either the N-terminus or C-terminus of *aipr-1* by CRISPR. *aipr-1(zw86)* mutants are hyperactive compared to the wild type and *aipr-1* nulls arrest at early larval stages. Both GFP insertion strains displayed normal locomotion, suggesting that the GFP insertion did not disrupt AIPR-1 function. We also performed electrophysiological analyses of the AIPR-1::GFP strain, and found that evoked responses and minis at the neuromuscular junction were similar to those of the wild type (Supplementary Fig. 6a, b). However, tagged endogenous AIPR-1 was too dim to be imaged along the nerve cords. To address the reviewer's concern of overexpression, we injected Prb-3::AIPR-1::GFP construct at a 10-fold lower concentration (2 ng/ μ l) and reimaged the strain coexpressing AIPR-1::GFP and TagRFP::ELKS-1 using shorter exposure times to minimize overexposure (Fig. 3d). The new result still support the notion that AIPR-1 is enriched at synaptic sites.

Comment 3

The characterization of motor neuron synapse development in *aipr-1* mutants described in Suppl Fig 2 needs to be more comprehensive. The authors need to use additional markers of pre-synapse (SYD-1, SNB-1) and post-synapse (UNC-29 and UNC-49). This is particularly important in light of the fact that *aipr-1* has an effect on synaptic vesicle biogenesis. To rule out the possibility of a developmental defect that could account for the increased neurotransmitter release in *aipr-1* mutants, the authors could apply pharmacological agonists and/or antagonists of RyR-1 only at adult stages.

Response

We performed several experiments to address this comment of the reviewer.

First, we expressed a presynaptic marker (GFP::ELKS-1) in wild-type and *aipr-1(zw86)* worms, and quantified the density of synapses in the dorsal nerve cord. We found that the density of GFP::ELKS-1 puncta was similar between these two strains (Supplementary Fig. 4d), which is in agreement with the RIM immunostaining result (Supplementary Fig. 4c).

p. 4. "The *aipr-1(zw86)* mutants do not exhibit morphological changes in acetylcholine and GABA motor neurons (Supplementary Fig. 4a, b) or an increase in the density of presynaptic sites (Supplementary Fig. 4c, d), suggesting hypersecretion is caused by a defect in synaptic function rather than development."

Second, we analyzed the expression and localization of UNC-29 (a key subunit of postsynaptic acetylcholine receptors in muscle) along the dorsal nerve cord utilizing a strain with UNC-29 labeled by TagRFP through single-copy insertion (EN208, kindly provided by Dr. Jean-Louis Bessereau, Pinan-Lucarre et al., *Nature* 2014). We found that UNC-29 expression and localization were similar between wild type and *aipr-1(zw86)* (Fig. 2e).

p. 4. “... the expression of a tagRFP-tagged UNC-29³³, which is a key subunit of muscle acetylcholine receptor³⁴, was similar between the wild type and *aipr-1(zw86)* (Fig. 2e), suggesting that muscle physiology is normal.”

Figure 2e Legend: “tagRFP-labeled muscle acetylcholine receptor (UNC-29) distribution and fluorescence intensity is normal in *aipr-1(zw86)* (wt, n = 20; *aipr-1*, n = 23, unpaired t-test). Scale bar, 10 μ m.”

Third, we directly applied acetylcholine and GABA to muscles and determined that there were no changes in neurotransmitter receptor density.

p. 4. “Furthermore, direct application of either acetylcholine or GABA onto muscle produces normal currents in *aipr-1(zw86)* (Fig. 2d),”

Finally, we examined the effect of ryanodine (100 μ M) on synaptic transmission in *aipr-1(zw86)*, and found that acute inhibition of ryanodine receptors greatly reduced both evoked responses and minis (Fig. 4a, b), suggesting that the increased neurotransmitter release in *aipr-1* mutants is not due to a secondary developmental defect.

p. 5. “Similarly, acute inhibition of ryanodine receptors using 100 μ M ryanodine abolished the increased evoked responses and minis in *aipr-1* mutants (Fig. 4a, b), suggesting that the suppression of defects in *aipr-1* by *ryr-1* mutations is not due to a developmental defect.”

Comment 4

More details on the transcriptional fusions for *ryr-1a* and *ryr-1b* are needed. Based on the well-established expression pattern of *ryr-1* in muscle, more details are needed for the two promoters used, especially because there is another gene really close and upstream of the *ryr-1* locus.

Response

The *ryr-1* gene encodes four splice isoforms (a, b, c, d) with two different initiation sites. In the original manuscript, muscle expression was observed with a GFP reporter construct containing 7-kb DNA upstream of the initiation site of *ryr-1a* and *ryr-1b* whereas neuronal expression was observed with a GFP reporter construct containing 4.8 kb DNA upstream of the initiation site of *ryr-1c* and *ryr-1d*. These two promoters were called *Pryr-1a* and *Pryr-1b* in the original manuscript. We now realize that these two names are confusing because it is difficult for readers to match them with the splice isoforms. To make things clearer, we have revised the legend of Supplementary Figure 7. In addition, we re-assessed *ryr-1* muscle expression using a shorter promoter sequence (2.5 kb), which did not include the upstream gene, to address the reviewer’s concern over the use of the 7-kb promoter. This new GFP transcriptional fusion also resulted in muscle-specific GFP expression (Supplementary Fig. 7a).

“Supplementary Figure 7. RYR-1 is expressed in muscles and neurons. The *ryr-1* gene encodes four alternatively spliced isoforms (a, b, c, d) with two different initiation sites (www.wormbase.org). **a**, A GFP reporter construct containing 2.5 kb sequence upstream of the initiation site of *ryr-1a* and *ryr-1b* (*Pryr-1_{a, b}::GFP*) drives expression in body-wall muscles (BWM) and vulva muscles (VM). **b**, A GFP reporter construct containing 4.8 kb sequence upstream of the initiation site of *ryr-1c* and *ryr-1d* (*Pryr-1_{c, d}::GFP*) drives expression in many neurons, including motor neurons in the ventral nerve cord (VNC). Scale bars, 20 μ m.”

Comment 5

On a similar note, it will be comforting to see what happens to amplitude and minis after muscle-specific RNAi for *ryr-1* in the *aipr-1* mutant.

Response

As the reviewer suggested, we tested whether muscle-specific *ryr-1* RNAi may alter neuromuscular transmission in the *aipr-1(zw86)* mutant. We found that neither evoked responses nor minis were affected by muscle-specific RNAi, suggesting that the synaptic phenotypes of the *aipr-1* mutant did not result from a loss of the ryanodine receptor in muscles. This conclusion is consistent with results of our previous study, which showed that synaptic transmission defects of a *ryr-1* null mutant can be rescued by expressing wild-type RYR-1 in neurons but not muscle cells (Liu et al., *J Neurosci* 25: 6745-6754, 2005).

p. 5. “Knockdown of *ryr-1* specifically in muscle had no effect on either evoked responses or minis of *aipr-1(zw86)* (Fig. 4a, b), suggesting that loss of presynaptic *ryr-1* occludes the synaptic phenotypes of the *aipr-1* mutant.”

Comment 6

The text (Results, line 137) is a bit inaccurate because it states that *C. elegans* *ryr-1* is only in the nervous system.

Response -- The error has been corrected.

p. 4. “Reporter constructs demonstrated that the upstream promoter is expressed in muscle; whereas the downstream promoter is expressed in neurons (Supplementary Fig. 7).”

Minor Points:

Comment - 1. At the beginning of the Results, it must be clearly stated that this is NMJ recordings, so that the reader gets familiar with the system at the very beginning of the text.

Response – Done.

p. 3. “The increased activity observed in the *slo-1(gf) aipr-1(zw86)* double mutant suggests that a potential physiological function of AIPR-1 is to limit synaptic activity. Indeed, electrophysiological recordings from the neuromuscular junction demonstrated that the *zw86* mutation increases synaptic transmission in *slo-1(gf)* mutants.”

Comment - 2. Suppl Fig 1 mixed up with Suppl Fig 3. Slo-1 localization in DNC seems affected. Better quantification is needed.

Response – We thank the reviewer for pointing out the mixing up of Suppl Figs 1 and 3, which has been corrected. As suggested, we quantified the density of SLO-1::GFP puncta in the dorsal nerve cord, and found that they were not altered in the *aipr-1(zw86)* mutant.

p. 2. “Moreover, *aipr-1(zw86)* does not alter either SLO-1 expression or subcellular localization (Supplementary Fig. 3c-e), unlike other mutants that suppress SLO-1(*gf*)^{28, 30, 31}.”

Comment - 3. Line 104. Mouse AIP rescued AIPR-1 expression. No data for this claim is provided.

Response – We actually did not examine the effect of mouse AIP on AIPR-1 expression. We apologize for this error, which has been corrected.

Comment - 4. How many times were the *airp-1(zw86)* mutants outcrossed before the comparison with *slo-1(gf)* mutants was performed?

Response - The *airp-1(zw86)* was outcrossed five times before the comparison.

p. 8. "The *zw86* mutant was outcrossed 5 times before analyses."

Comment - 5. Line 188 and Fig 6b: the authors do not provide an explanation as to why the *airp-1;ryr-1* double mutants has a remarkable increase in synaptic vesicles compared to single mutants.

Response – Supplementary Figure 10 Legend: "The increase in synaptic vesicles is profound in the *airp-1 ryr-1* double mutant in acetylcholine neurons; it is not clear whether this is due to the small data set possible by EM, to a synthetic defect of the mutations, or to a specific response of the acetylcholine synapses."

Comment - 6. Line 209: More than one target is also possible.

Response – We agree with the reviewer.

p. 6. "In addition to the ryanodine receptor, AIP must act on at least one other target that limits synaptic vesicle number at neuromuscular junctions."

REVIEWERS' COMMENTS:

Reviewer #3 (Remarks to the Author):

The authors have satisfactorily addressed all my concerns and questions.

Reviewer #4 (Remarks to the Author):

All my concerns have been addressed in the revised version of the manuscript. I believe the revised form is now suitable for publication in Nat Coms.